



# Roles of pH, ionic strength, and sulfate in the aqueous nitrate-mediated photooxidation of green leaf volatiles

Yuting Lyu[1,2], Taekyu Joo[3], Ruihan Ma[1], Mark Kristan Espejo Cabello[1,2], Tianye Zhou[1], Shun Yeung[1], Cheuk Ki Wong[1], Yifang Gu[1], Yiming Qin[1], Theodora Nah[1,2]*

[1]*School of Energy and Environment, City University of Hong Kong, Hong Kong SAR, China*
[2]*State Key Laboratory of Marine Pollution, City University of Hong Kong, Hong Kong SAR, China*
[3]*Department of Earth and Environmental Sciences, Korea University, Seoul, South Korea*

*Correspondence to*: Theodora Nah (theodora.nah@cityu.edu.hk)

**Abstract.** Biotic and abiotic stresses can lead to terrestrial green plants releasing green leaf volatiles (GLVs), which can partition into atmospheric aqueous phases where they can undergo oxidation to form aqueous secondary organic aerosols (aqSOA). Anthropogenic emission changes have resulted in nitrate becoming an increasingly important component of atmospheric aqueous phases, which has significant implications for aqSOA formation since nitrate photolysis produces oxidants. Nevertheless, sulfate remains the main inorganic aqueous component in most regions, and thus controls the pH and

ionic strength of atmospheric aqueous phases. We report results from laboratory investigations of the effects of pH, ionic strength, and sulfate on the reaction kinetics and aqSOA formation of the aqueous nitrate-mediated photooxidation of four GLVs, *cis*-3-hexen-1-ol, *trans*-2-hexen-1-ol, *trans*-2-penten-1-ol, and 2-methyl-3-buten-2-ol. Our results showed that the aqueous reaction medium conditions, i.e., dilute cloud/fog vs. concentrated aqueous aerosol conditions, governed the effects that pH, ionic strength, and sulfate have on the GLV degradation rates and aqSOA mass yields. Most notably, reactions

initiated by sulfate photolysis will have significant effects on the GLV degradation rates and aqSOA mass yields in aqueous aerosols, but not in cloud/fog droplets. In addition to providing new insights into aqSOA formation from the aqueous reactions of GLVs in regions with substantial concentrations of nitrate in cloud, fog, and aqueous aerosols, this study highlights how nitrate and sulfate photochemistries can couple together to influence the reactions of water-soluble organic compounds and their aqSOA formation in aqueous aerosols, which have implications for our evaluations of aqueous organic

aerosol lifetimes and composition.

## 1 Introduction

Biogenic volatile organic compounds (BVOCs) contribute more than 80 % of the global volatile organic compound (VOC) emissions (Guenther et al., 2012; Sindelarova et al., 2014). Isoprene and monoterpenes comprise more than half of the total annual BVOC emissions (Sindelarova et al., 2014). The remaining BVOCs include green leaf volatiles (GLVs),

which are $C_5$ to $C_6$ unsaturated organic compounds with aldehyde, alcohol, or ester functional groups (Sarang et al., 2021a). GLVs are emitted during the decomposition of $C_{18}$ polyunsaturated fatty acids in leaves when vegetation is exposed to

 

herbivores, pathogens, or harsh weather conditions (Ameye et al., 2018; Matsui and Engelberth, 2022a; Silva et al., 2021). They are also emitted by cyanobacteria and algae during bloom events (García-Plazaola et al., 2017). GLVs have the potential to contribute substantially to the local secondary organic aerosol (SOA) budget due to their increased emissions

when vegetation is subjected to biotic and abiotic stresses. A previous study reported that GLV emissions from Amazon tropical forests increased significantly in the afternoon due to the plants' response to rising temperatures, whereas isoprene and monoterpene emissions decreased (Jardine et al., 2015). Global GLV emissions will potentially increase in the future due to climate change and the increasing use of new fumigation-based agricultural, horticultural, and forestry practices (Cofer et al., 2018; Matsui and Engelberth, 2022b; Su et al., 2020). Thus, GLVs may play increasingly important roles in

atmospheric chemistry, which necessitates increasing our knowledge of their multiphase reactions and SOA formation.

GLVs can be oxidized by ozone and hydroxyl radicals ($\cdot$OH) in the gas phase to produce low volatility products, with reported SOA mass yields ranging from 0.7 to 20 % (Hamilton et al., 2009; Harvey et al., 2014; Jaoui et al., 2012; Mentel et al., 2013). Due to their moderately high water solubilities, GLVs can dissolve into atmospheric aqueous phases (e.g., aqueous aerosols, cloud and fog droplets), where they can be oxidized by aqueous oxidants such as $\cdot$OH, sulfate anion

radicals ($SO_4^{-}\cdot$), nitrate radicals ($NO_3\cdot$), triplet organic excited states ($^3C^*$), and singlet oxygen ($^1O_2^*$) (Richards-Henderson et al., 2014; Richards-Henderson et al., 2015; Sarang et al., 2021a; Sarang et al., 2021b; Sarang et al., 2023). Higher quantities of low volatility products are formed from aqueous reactions compared to gas-phase reactions, with previous studies reporting aqueous SOA (aqSOA) mass yields as high as 88 % though this will depend on the GLV and the aqueous oxidant (Richards-Henderson et al., 2014; Richards-Henderson et al., 2015). However, these previous studies were mostly conducted

under dilute aqueous conditions mimicking aqueous cloud/fog droplets. Differences in the physicochemical properties of the aqueous reaction medium will impact reaction rates in cloud/fog droplets vs. aqueous aerosols (Herrmann et al., 2015), but little is currently known about the aqueous oxidation of GLVs under more concentrated aqueous aerosol-like conditions.

The liquid water concentrations (LWC) of cloud and fog droplets typically fall in the range of 0.05 to 0.5 g/m$^3$ (Achtert et al., 2020; Kim et al., 2022; Korolev et al., 2007), whereas the LWC of aqueous aerosols fall in the range of $10^{-7}$ to

$10^{-3}$ g/m$^3$ (Herrmann et al., 2015). Thus, the concentrations of dissolved organic and inorganic compounds are higher in aqueous aerosols, with their dry masses close to the liquid water mass (Nguyen et al., 2016). The concentrations of inorganic ions, particularly nitrate and sulfate, primarily govern the acidities and ionic strengths of atmospheric aqueous phases. Cloud and fog droplets (pH 2 to 7) are generally less acidic than aqueous aerosols (pH 0 to 5) due to the more frequent ammonia dissolution and higher buffering capacities of cloud and fog droplets (Pye et al., 2020; Tilgner et al., 2021). The ionic

strengths of atmospheric aqueous phases span a large range ($10^{-5}$ M to $10^0$ M), with the ionic strengths of aqueous aerosols being several orders of magnitude higher than those of cloud and fog droplets (Herrmann et al., 2015; Tilgner et al., 2021). Under the high ionic strength conditions in aqueous aerosols, substantial ion association occurs, which will affect the activity coefficients of organic compounds, resulting in reactions occurring under non-deal conditions (Herrmann, 2003). Previous studies have reported that ionic strength significantly affect the aqueous reactions of some organic compounds and



subsequent product formation (Herrmann, 2003; Mekic et al., 2018a; Mekic et al., 2018b; Mekic and Gligorovski, 2021; Zhou et al., 2019). In addition to contributing to the acidity and ionic strength of atmospheric aqueous phases, inorganic nitrate and sulfate can undergo photolysis to produce various reactive species that react with organic compounds. The tropospheric irradiation of nitrate in atmospheric aqueous phases is known to produce aqueous reactive species such as ·OH, nitric oxide radicals (NO·), and nitrogen dioxide radicals (NO$_2$·) that can react with organic compounds (Mack and Bolton,

1999; Gen et al., 2022). Even though a recent study reported that sulfur containing radicals (e.g., SO$_4$·$^-$) are formed during the tropospheric irradiation of aqueous sulfate aerosols (Cope et al., 2022), the mechanisms behind their formation are still not well understood. While nitrate is increasingly important in regions with large reductions in sulfur dioxide emissions and/or with high ammonia emissions (Heald et al., 2012; Schaap et al., 2004; West et al., 1999), sulfate remains the dominant inorganic constituent of atmospheric aqueous phases in most regions (Bianco et al., 2020). At present, little is

known about how inorganic nitrate and sulfate salts influence the aqueous oxidation of GLVs.

        In this study, we investigated the nitrate-mediated photooxidation of four GLVs, *cis*-3-hexen-1-ol, *trans*-2-hexen-1-ol, *trans*-2-penten-1-ol, and 2-methyl-3-buten-2-ol (Figure 1), under cloud/fog-like and aqueous aerosol-like conditions. These four GLVs, which are amongst some of the more abundant GLVs, have Henry's law constants between 60 to 120 M atm$^{-1}$ (Sander, 2023; Sarang et al., 2021a). Thus, they can partition efficiently into cloud/fog droplets and moderately into

aqueous aerosols (Figure S1). We investigated the effects of pH, ionic strength, and sulfate on the reaction kinetics and aqSOA mass yields under cloud/fog-like and aqueous aerosol-like conditions. Results from this study provide new insights into the aqueous photooxidation of GLVs in regions with substantial levels of nitrate in cloud and fog droplets and aqueous aerosols, and more generally, how sulfate photochemistry impacts the aqueous nitrate-mediated photooxidation of other water-soluble organic compounds.

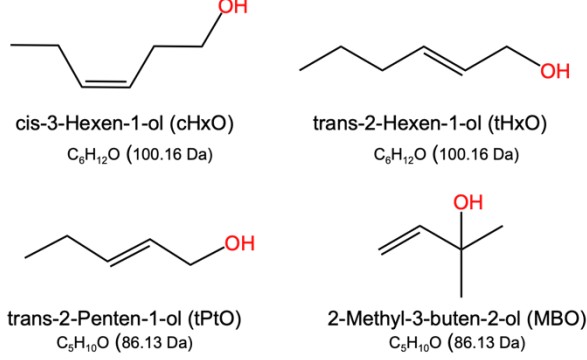


**Figure 1.** The four model GLVs used in this study.



## 2 Methods

### 2.1 Chemicals and Solutions

All the chemicals were used as received. *cis*-3-Hexen-1-ol (cHxO, 98 %), *trans*-2-hexen-1-ol (tHxO, 96 %), 2-
methyl-3-buten-2-ol (MBO, 97 %), benzoic acid (BA, 99.5 %) and p-hydroxybenzoic acid (pHBA, 99 %) were purchased
from J&K Scientific. *Trans*-2-penten-1-ol (tPtO, ≥ 95 %) was purchased from Aladdin Scientific. Ammonium nitrate
($NH_4NO_3$, ≥ 95 %) and ammonium sulfate (($NH_4$)$_2SO_4$, 99+ %) was purchased from Fisher Scientific. Sulfuric acid ($H_2SO_4$,
%) was purchased from VWR Chemicals BDH®. Milli-Q ultrapure water (18.2 MΩ cm) was used to prepare all solutions.

Table 1 shows the concentrations of solutions used to simulate cloud/fog-like and aqueous aerosol-like conditions in
photochemistry experiments. The same nitrate/GLV molar concentration ratio (i.e., 2.5:1) was used for both cloud/fog-like
and aqueous aerosol-like conditions. The concentrations of the GLVs and $NH_4NO_3$ were set to be 100 times higher for the
aqueous aerosol-like conditions than the cloud/fog-like conditions. The nitrate concentration (i.e., 250 µM) used for
cloud/fog-like conditions is close to those measured in cloud water in Hong Kong (average of 238 µM) (Li et al., 2020) and
is within the global range for nitrate in continental cloud water (Bianco et al., 2020). Even though the nitrate concentration in
aqueous aerosols could reach molar levels (Bzdek et al., 2020), we used 25 mM of $NH_4NO_3$ in experiments that simulate
aerosol-like conditions to maintain the same nitrate/GLV molar concentration ratio as in experiments that simulate
cloud/fog-like conditions. The pH of unbuffered solutions (no addition of $H_2SO_4$) were close to 5, and it was selected as the
higher bound to study pH the effects on the nitrate-mediated photooxidation of GLVs. The lower bound of pH was set to 3
through the addition of $H_2SO_4$. (($NH_4$)$_2SO_4$ was added into the solutions to control the sulfate concentrations in and ionic
strengths of the solutions. Since the acid dissociation constant (p$K_a$) for $HSO_4^- \leftrightarrow H^+ + SO_4^{2-}$ is around 2.0 at 25 °C (Dickson
et al., 1990), $SO_4^{2-}$ are expected to be the dominant species even at the lower bound of pH 3. The pH (i.e., pH 3 vs. 5) used in
this study fall within the ranges for cloud and fog droplets and aqueous aerosols (Herrmann et al., 2015; Pye et al., 2020;
Tilgner et al., 2021). The ionic strengths ($I_{total}$) of the solutions were calculated using the following equation:

$$I_{total} = \frac{1}{2} \times \sum_{i=1}^{i=n} c_i z_i \tag{1}$$

where $c_i$ and $z_i$ are the concentration (M) and charge, respectively, of inorganic ion $i$ for $H^+$, $NH_4^+$, $NO_3^-$, and $SO_4^{2-}$. The ionic
strengths used in this study, i.e., 0.002 M vs. 0.02 M for cloud/fog-like conditions and 0.5 M vs. 3.3 M for aqueous aerosol-
like conditions, fall within the ranges for ionic strengths for clouds/fog droplets and continental aerosols (Herrmann et al.,
2015). The addition of GLVs, (($NH_4$)$_2SO_4$, and $H_2SO_4$ did not alter the molar absorptivity of $NH_4NO_3$ (Figure S2).



**Table 1.** Concentrations of GLVs, NH$_4$NO$_3$, H$_2$SO$_4$, (NH$_4$)$_2$SO$_4$, pH, and $I_{total}$ of solutions used to simulate cloud/fog-like and aqueous aerosol-like conditions in photochemistry experiments.

| Simulated condition | [GLV] (mM) | [NH$_4$NO$_3$] (mM) | [H$_2$SO$_4$] (mM) | pH | [(NH$_4$)$_2$SO$_4$] (mM) | $I_{total}$ (M) |
|---|---|---|---|---|---|---|
| Cloud/fog-like | 0.1 | 0.25 | 0.5 | 3 | 0.135 | 0.002 |
| | | | | | 6.135 | 0.02 |
| | | | 0 | 5 | 0.583 | 0.002 |
| | | | | | 6.580 | 0.02 |
| Aqueous aerosol-like | 10 | 25 | 0.5 | 3 | 158 | 0.5 |
| | | | | | 1085 | 3.3 |
| | | | 0 | 5 | 158 | 0.5 |
| | | | | | 1085 | 3.3 |

**2.2 Photochemistry Experiments**

The solutions were placed into open cylindrical quartz tubes (1.2 cm inner diameter), which were placed on a rotating sample holder located in the middle of a photoreactor (Rayonet RPR-200, Southern New England UV Co.) surrounded by UVB lamps (RPR-3000Å, Southern New England UV Co.). The photon flux in the photoreactor ranged from 275 to 400 nm and peaked at 311 nm (Figure S2). The temperature inside the photoreactor during experiments was maintained at around 30 ℃ by a cooling fan located at the bottom of the photoreactor. Aliquots of 1 mL and 0.1 mL were extracted from the illuminated solutions at different reaction times for offline chemical analysis during experiments simulating cloud/fog-like and aqueous aerosol-like conditions, respectively. For experiments simulating aqueous aerosol-like conditions, the extracted volume (0.1 mL) was diluted with Milli-Q ultrapure water by a factor of 10 prior to chemical analysis to ensure that the measured signals stayed within the linear detection range of detector. The decays of the GLVs were measured using an ultrahigh-performance liquid chromatography coupled to a photodiode array detector (UPLC-PDA, H-class, Waters). A Kinetex Polar C18 column (2.6 μm, 100 × 2.1 mm) equipped with a security guard and Polar C18 pre-column was used for the measurement of the four GLVs. An isocratic elution program set to 0.3 mL/min was used. The mobile phases of water and acetonitrile were run at a ratio of 80:20 for cHxO, tHxO, and MBO, and at a ratio of 85:15 for tPtO. The detection wavelengths were set to 240 nm for MBO, and 210 nm for the other three GLVs. All the experiments and measurements were performed three times. All the decays of the GLVs followed apparent first order reaction kinetics reasonably well, thus they were fitted with the following equation:



$$\ln\left(\frac{[GLV]_t}{[GLV]_0}\right) = -k_{obs}t \tag{2}$$

where $k_{obs}$ is the pseudo-first order rate obtained from the exponential fit to the photodegradation of the GLV, and $[GLV]_t$ and $[GLV]_0$ are the concentrations of individual GLV measured by UPLC-PDA at illumination times $t$ and 0, respectively.

No loss in GLVs were observed in dark control experiments conducted in the absence and presence of nitrate and sulfate. Only MBO showed some loss during illumination in control experiments conducted in the absence of nitrate and sulfate. This was surprising since the four GLVs were not expected to undergo direct photolysis as they do not absorb light at
wavelengths larger than 280 nm (Richards-Henderson et al., 2014; Sarang et al., 2021a). The observed loss of MBO could be due to MBO evaporation since it has a higher vapor pressure ($3.08 \times 10^{-2}$ atm) compared to the other three GLVs (cHxO: $1.23 \times 10^{-3}$ atm, tHxO: $1.20 \times 10^{-3}$ atm, and tPtO: $3.46 \times 10^{-3}$ atm) based on estimations using EPI Suite™ (Epa, 2024). The $k_{obs}$ values measured for the MBO decays in nitrate-mediated photooxidation experiments were subsequently corrected by subtracting the MBO loss rates from control experiments conducted in the absence of nitrate and sulfate.

To gain insights into how concentration, pH, and ionic strength affect ·OH formation during nitrate-mediated photooxidation, a separate set of experiments (i.e., GLVs were not present in the solutions) using BA (10 µM) as the ·OH probe compound was conducted to estimate the steady-state concentrations of ·OH ($[·OH]_{ss}$) using the same methodology as our past studies (Lyu et al., 2023; Yang et al., 2021; Yang et al., 2023). p-HBA, which is formed from the reaction of ·OH with BA ($k_{BA+OH} = 5.9 \times 10^9$ M$^{-1}$ s$^{-1}$ (Herrmann et al., 2010)) at a yield of 0.17 (Anastasio and Mcgregor, 2001), was
measured in these experiments using an ultra-high performance liquid chromatography system (1290 system, Agilent) coupled to a high-resolution quadrupole-time-of-flight mass spectrometer (X500R QTOF MS/MS, Sciex) (UPLC-MS) equipped with an electrospray ionization (ESI) source that was operated in negative mode (Section S1). Solid phase extraction (SPE) using SPE cartridges (Oasis MAX, 60 mg, 3 cc, 60 µm, Waters) was performed before UPLC-MS analysis to remove inorganic salts from the before UPLC-MS analysis. However, due to the high concentrations of inorganic salts in
the solutions used to simulate aqueous aerosol-like conditions, we were unable to completely remove the inorganic salts from these samples. Thus, we only investigated the effects of concentration, pH, and ionic strength on $[·OH]_{ss}$ for the cloud/fog-like conditions.

## 2.3 AqSOA Mass Yields

Aerosol mass spectrometry was used to measure the aqSOA mass yields of the four GLVs, Aliquots of 10 mL and 1
mL were extracted from the illuminated solutions at one GLV lifetime (i.e., $\tau = \frac{1}{k_{obs}}$, when 37 % of the initial concentration of the GLV remained) in experiments simulating cloud/fog-like and aqueous aerosol-like conditions, respectively. The time points equivalent to one GLV lifetime were determined from the forementioned kinetic experiments. For experiments



simulating aqueous aerosol-like conditions, the extracted volume (1 mL) was diluted with Milli-Q ultrapure water by a factor of 300 prior to aerosol mass spectrometry analysis to ensure that the measured signals stayed within the linear detection range of detector. Each sample solution was injected at 10 mL/h by a syringe pump (Model 100, KD Scientific) into an aerosol generation system (Model 9200, Brechtel Manufacturing Incorporated), which aerosolized the solution. The atomizer system used nitrogen gas (99.999 % purity) as the carrier gas at a flow rate of 4.5 L/min. The aerosols generated were passed through an inline dryer before entering a time-of-flight aerosol chemical speciation monitor (ACSM, Aerodyne Research Inc.). All the experiments and measurements were performed three times.

The aqSOA mass yield ($Y_{SOA}$) was calculated using the following equation (Jiang et al., 2023; Ma et al., 2021):

$$Y_{SOA} = \frac{Organic\ mass\ increased}{GLV\ mass\ decreased} = \frac{[Org]_\tau - [Org]_0}{[GLV]_0 - [GLV]_\tau}$$

$$= \frac{[Org]_{ACSM,\tau} \times \frac{[SO_4^{2-}]_\tau}{[SO_4^{2-}]_{ACSM,\tau}} - [Org]_{ACSM,0} \times \frac{[SO_4^{2-}]_0}{[SO_4^{2-}]_{ACSM,0}}}{(1 - 0.37) \times [GLV]_0} \quad (3)$$

where the [Org], [GLV], and [$SO_4^{2-}$] were the concentrations of organics, the GLV of interest, and sulfate, respectively. The terms with subscript of ACSM indicate the mass concentrations (in mg/L) of aerosols measured by the ACSM, which is different from the molar concentrations (mol/L) of solutions which do not have the ACSM subscripts. The subscripts $\tau$ and 0 indicate the sample solutions obtained at one GLV lifetime and before illumination, respectively. The concentrations of sulfate in the solutions before and after illumination were assumed to be the same (i.e., [$SO_4^{2-}$]$_\tau$ = [$SO_4^{2-}$]$_0$). The concentration of sulfate in each sample solution (Table 1) was used as the internal standard to scale the concentrations of organics measured by ACSM to those of the solutions. This is because sulfate is non-refractory and is expected to be collected and quantified by the ACSM, which had a capture vaporizer with a collection efficiency of 1 (Daellenbach et al., 2016; Xu et al., 2018; Joo et al., 2021). Additionally, the sulfate and organic composition in the atomized aerosols are expected to be internally mixed together (Ma et al., 2021).

## 3 Results and Discussion

### 3.1 Cloud/fog-like Conditions

### 3.1.1 Reaction Kinetics

The concentrations of the four GLVs decreased upon irradiation in the presence of nitrate and sulfate. In contrast to MBO, no decays were observed for cHxO, tHxO, and tPtO when only sulfate was present (but not nitrate) in the solutions. In the absence of nitrate, the MBO decay rates obtained in the absence and presence of sulfate were similar. The MBO decay in the absence of nitrate could be due to MBO evaporation since it has a higher vapor pressure than the other three GLVs.





Overall, these results indicated that sulfate has an insignificant effect on the kinetics of the four GLVs under cloud/fog-like
conditions.

Figure 2 shows the $k_{obs}$ values for the four GLVs upon irradiation in the presence of nitrate at different pH (i.e., 3 vs.
5) and ionic strength (i.e., 0.002 M vs. 0.02 M). Separate experiments performed in the absence of GLVs and using BA as
the $\cdot$OH probe compound showed that the estimated $[\cdot OH]_{ss}$ values decreased with pH under these cloud/fog-like conditions
(Figure S3), consistent with results reported by Lyu et al. (2023). The $k_{obs}$ values for the four GLVs were on the orders of $10^{-5}$
to $10^{-4}$ s$^{-1}$ for the four GLVs. The decays in the four GLVs upon irradiation in the presence of nitrate were due to the
reactions of the GLVs with reactive species produced from nitrate photolysis such as $\cdot$OH, NO$\cdot$, and NO$_2\cdot$ (Table S1). Even
though approximately equal quantities of $\cdot$OH and NO$_2\cdot$ are produced during nitrate photolysis (Chen et al., 2019; Zhang et
al., 2021), the typical reactivities of NO$_2\cdot$ are 2 to 5 orders of magnitude lower than $\cdot$OH (Chen et al., 2019; Ford et al., 2002;
Zhang et al., 2021). Other reactive species produced during nitrate photolysis (e.g., hydroperoxide radicals (HO$_2\cdot$) and
superoxide ions (O$_2^-\cdot$)) are also expected to have lower reactivities compared to $\cdot$OH (Bielski et al., 1985; Mack and Bolton,
1999). Thus, the decays of the GLVs were likely governed mostly by their reactions with $\cdot$OH, though contributions from
their reactions with reactive species other than $\cdot$OH cannot be discounted.

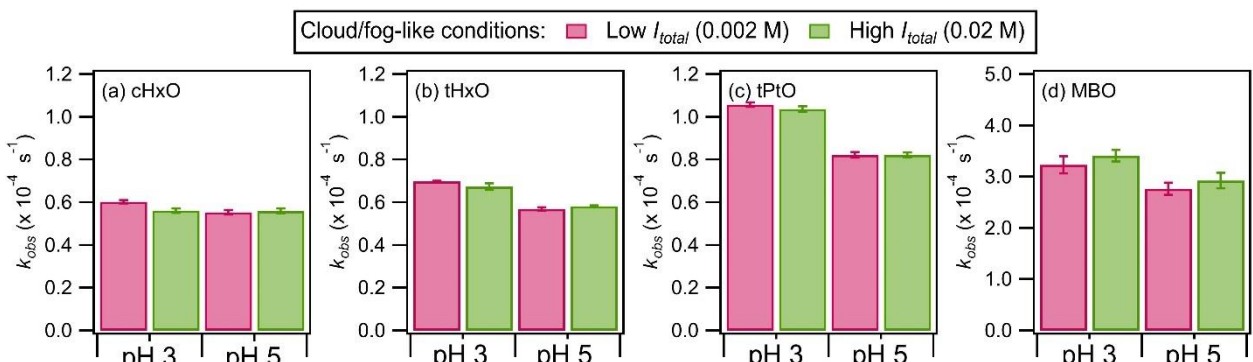

**Figure 2.** The $k_{obs}$ values for the four GLVs during nitrate-mediated photooxidation under cloud/fog-like conditions (Table
1). The error bars represent one standard deviation originating from triplicate experiments and measurements. Statistical
analyses (student's $t$ test) on the differences in the $k_{obs}$ values are presented in Tables S2 to S5.

Under the same ionic strength conditions, the four GLVs had higher $k_{obs}$ at pH 3 than at pH 5, though the pH-
dependent trends for cHxO at $I_{total}$ = 0.002 M and 0.02 M were statistically insignificant ($p > 0.05$) (Table S2). The four
GLVs do not have acidic H atoms, thus they do not undergo acid dissociation to form different relative abundances of
deprotonated and neutral forms with different reactivities at different pH. Additionally, Richard-Henderson et al. (2014)
showed that the $\cdot$OH rate constants for many GLVs do not depend on pH. Hence, the pH-dependent $k_{obs}$ trends in Figure 2
were due to the pH-dependent formation of $\cdot$OH (Figure S3) and other reactive species from nitrate photolysis. HNO$_2$, whose





production from nitrate photolysis is favored over $NO_2^-$ production at pH $\leq 3.5$ (Marussi and Vione, 2021), has a higher quantum yield for ·OH formation than $NO_2^-$ in the near-UV region (Arakaki et al., 1999; Marussi and Vione, 2021). Thus,

the formation rates and concentrations of ·OH produced at pH 3 are higher than at pH 5, which would explain the higher $k_{obs}$ at pH 3. There were no statistically significant differences in the $k_{obs}$ values for $I_{total}$ of 0.002 M vs. 0.02 M under the same pH conditions for the four GLVs ($p > 0.05$). This indicated that ionic strength (and sulfate) has an insignificant effect on the reaction kinetics of the four GLVs under cloud/fog-like conditions. Overall, only pH impacted the reaction kinetics of the four GLVs significantly under cloud/fog-like conditions.

**3.1.2 AqSOA Mass Yields**

Figure 3 shows the $Y_{SOA}$ values for the four GLVs measured at one GLV lifetime during irradiation in the presence of nitrate at different pH (i.e., 3 vs. 5) and ionic strength (i.e., 0.002 M vs. 0.02 M). The measured $Y_{SOA}$ values (0 to 53 %) are in line with the range of $Y_{SOA}$ values (10 to 88 %) measured by Richards-Henderson et al. (2014) for five GLVs (including cHxO and MBO) at one GLV halftime in their reactions with ·OH generated from $H_2O_2$ photolysis under

cloud/fog-like conditions.

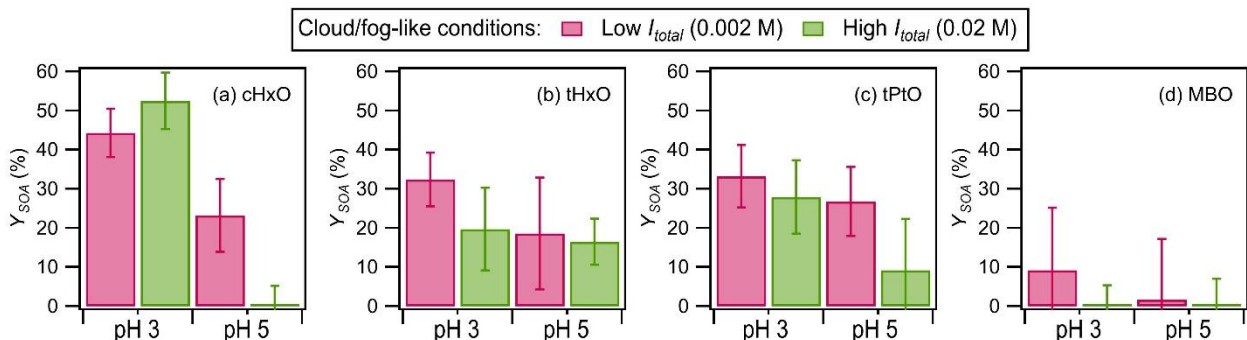

**Figure 3.** $Y_{SOA}$ values for the four GLVs at one GLV lifetime during nitrate-mediated photooxidation under cloud/fog-like conditions. The error bars represent one standard deviation originating from triplicate experiments and measurements, and include errors propagated from the standard deviations of the sulfate concentrations measured by ACSM. The ACSM-

measured organic signals for cHxO at pH 5 and $I_{total} = 0.02$ M, and for MBO at pH 3 and 5 at $I_{total} = 0.02$ M, and at pH 5 at $I_{total} = 0.002$ M were very low, resulting in close to zero organic concentrations and $Y_{SOA}$ values. Therefore, there was essentially no formation of low volatility products from these three experiments, and the $Y_{SOA}$ values were essentially zero. Statistical analyses (student's $t$ test) on the differences in the $Y_{SOA}$ values are presented in Tables S6 to S9.

Given the high reactivity of ·OH, reactions of the GLVs with ·OH are expected to produce products that contribute

substantially to aqSOA formation. However, we cannot discount the possibility that other reactive species produced from nitrate/sulfate photolysis also contributed to aqSOA formation. The reaction of ·OH with the GLV is expected to occur either



by ·OH addition to the C=C bonds to form hydroxy alkyl radicals, or by H abstraction from the C-H or O-H bonds to form alkyl radicals (Figures S4 and S5). The H bond dissociation energies at the $CH_2$, $CH_3$, and OH groups are around 393, 419, and 436 kJ mol$^{-1}$, respectively (Benson, 1976). H abstraction is expected to occur preferentially at C-H sites α to the OH group (Cooper et al., 2009; Sarang et al., 2023). The hydroxy alkyl radicals and alkyl radicals subsequently react with $O_2$ to form peroxy radicals ($RO_2$·), which then react with other $RO_2$· to form either higher molecular weight carbonyls and alcohols or alkoxyl radicals (RO·). RO· can undergo fragmentation reactions to form lower molecular weight compounds. Even though they are not shown in our proposed reaction mechanisms in Figures S4 and S5, bimolecular combination reactions involving $RO_2$· and RO· (e.g., $RO_2$· + $RO_2$·) that form oligomers could have also contributed to aqSOA formation.

Attempts to identify prominent low volatility products (and their formation pathways) that contributed to aqSOA using liquid chromatography-mass spectrometry were unsuccessful due to the presence of large quantities of inorganic salts in the samples, which negatively impacted the ionization efficiencies of the products. Nevertheless, low volatility products from both the ·OH addition and H abstraction channels likely contributed to aqSOA formation for the four GLVs. Sarang et al. (2023) previously detected products formed from both channels in their study of the aqueous ·OH oxidation of various GLVs. In the case of cHxO, the dominant products from the H abstraction channel were reported to be at least 15 kcal mol$^{-1}$ more stable than the products from the ·OH addition channel (Sarang et al., 2023). Subsequent density functional theory calculations indicated that both channels were important contributors to product formation due to the barrierless pathway in the ·OH addition channel and the formation of thermodynamically stable allylic alkyl radicals in the H abstraction channel (Sarang et al., 2023). Allylic alkyl radicals from the H abstraction channel similarly play important roles in product formation in the ·OH oxidation of large unsaturated organic compounds (Nah et al., 2014).

In contrast to cHxO, tHxO, and tPtO, the $Y_{SOA}$ values for MBO were not statistically different ($p > 0.05$) from 0 %. The substantial differences in the $Y_{SOA}$ values for MBO vs. cHxO, tHxO, and tPtO could be attributed to the molecular structure of MBO. MBO contains a terminal C=C bond that is adjacent to its OH group, whereas cHxO, tHxO, and tPtO contain non-terminal C=C bonds that are non-adjacent to their OH groups (Figure 1). Due to the molecular structure of MBO, the formation of RO· with oxygen radical centers adjacent to at least one OH functional group is enhanced for both the ·OH addition and H abstraction channels (Figure S5). Their close proximity to oxygenated functional groups increase the susceptibility of these RO· to fragmentation (Atkinson, 1997), which forms lower molecular weight compounds that may volatilize into the gas phase. Thus, the enhanced formation of RO· that preferentially fragment into higher volatility products during the reaction of MBO with ·OH under cloud/fog-like conditions would explain its low $Y_{SOA}$ values. In contrast, RO· formation (and thus, fragmentation) is not enhanced in the ·OH reactions of cHxO, tHxO, and tPtO due to the formation of primary and secondary $RO_2$· formed from the ·OH addition and H abstraction channels.

Under the same ionic strength conditions, the $Y_{SOA}$ values for the four GLVs were generally higher at pH 3 than at pH 5, though these pH-dependent trends were not statistically significant ($p > 0.05$) in some instances (Tables S6 to S9). It is



possible that the enhanced aqSOA formation at lower pH was due to acid-catalyzed reactions. The aqueous reaction of ·OH
with the four GLVs likely form various higher and lower molecular weight carbonyls (Figures S4 and S5) (Sarang et al.,
2021a; Sarang et al., 2023). Some of these carbonyls could have undergone acid-catalyzed reactions (e.g., hydration,
polymerization, aldol condensation) to form low volatility products (Ervens et al., 2011; Hallquist et al., 2009; Jang et al.,
2002). Additionally, it is possible that NO· enhanced the formation of low volatility products that contributed to aqSOA at
lower pH, possibly through the formation of low volatility organonitrates via the $RO_2·$ + NO· → $RONO_2$ pathway (Atkinson
and Arey, 2003). This is because NO· formation from nitrate photolysis would be enhanced at pH 3 (Table S1). $HNO_2$ is
favored over its conjugated base $NO_2^-$ at pH < 3.3 (Marussi and Vione, 2021). The production of NO· from the photolysis of
$HNO_2$ (R15 in Table S1) is nearly one order of magnitude faster than its formation from $NO_2^-$ photolysis (R10 in Table S1).

Interestingly, with the exception of cHxO at pH 3, the $Y_{SOA}$ values for the four GLVs were higher at $I_{total}$ = 0.002 M
than at $I_{total}$ = 0.02 M under the same pH conditions, though these ionic strength-dependent trends were not statistically
significant ($p > 0.05$) in some instances (Tables S6 to S9). This is in contrast to the insignificant effect that ionic strength had
on $k_{obs}$ (Figure 2). $(NH_4)_2SO_4$ was used to control the ionic strengths of the solutions (Table 1). Thus, our results indicated
that even though ionic strength and/or sulfate concentration had insignificant effects on the reaction kinetics under cloud/fog-
like conditions, they could significantly affect the formation of low volatility products. The lower $Y_{SOA}$ values measured at
higher $I_{total}$ and sulfate concentrations under the same pH conditions implied that fragmentation reactions that form volatile
lower molecular weight products were enhanced at higher $I_{total}$ and/or sulfate concentrations. Additionally, the higher $I_{total}$
conditions could have enhanced the partitioning of products to the gas phase due to the salting out effect (Peng and Wan,
1998).

### 3.2 Aqueous aerosol-like Conditions

### 3.2.1 Reaction Kinetics

Figure 4 shows the $k_{obs}$ values for the four GLVs upon irradiation in the presence of nitrate at different pH (i.e., 3 vs.
5) and ionic strength (i.e., 0.5 M vs. 3.3 M). The concentrations of the GLVs and $NH_4NO_3$ used in this set of experiments to
simulate aqueous aerosol-like conditions were both 100 times higher than those used to simulate cloud/fog-like conditions
while maintaining the same nitrate/GLV molar concentration ratio of 2.5:1 (Table 1). Even though the [·OH]$_{ss}$ values could
not be estimated for the aqueous aerosol-like conditions since the inorganic salts could not be completely removed prior to
UPLC-MS analysis, the [·OH]$_{ss}$ values were likely higher than those for the cloud/fog-like conditions. The $k_{obs}$ values for the
four GLVs were on the orders of $10^{-6}$ to $10^{-3}$ s$^{-1}$.





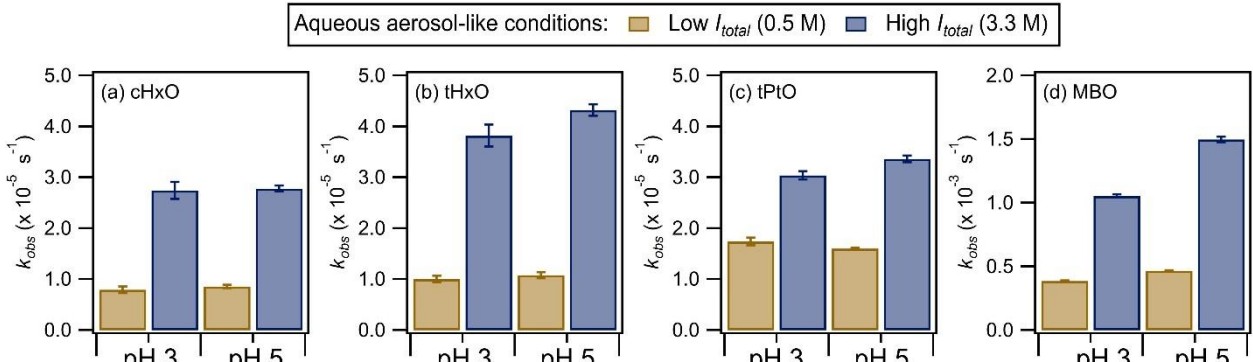

**Figure 4.** The $k_{obs}$ values for the four GLVs during nitrate-mediated photooxidation under aqueous aerosol-like conditions (Table 1). The error bars represent one standard deviation originating from triplicate experiments and measurements. Statistical analyses (student's $t$ test) on the differences in the $k_{obs}$ values are presented in Tables S10 to S13.

The $k_{obs}$ values measured for cHxO, tHxO, and tPtO under aqueous aerosol-like conditions were factors of 1.1 to 6.1 lower than those measured for cloud/fog-like conditions (Figure 2). The lower $k_{obs}$ values measured for these three GLVs under aqueous aerosol-like conditions could be due, in part, to the exponential decrease in the nitrate photolysis rate with increasing nitrate concentration (Ye et al., 2017). Consequently, the production of ·OH and other reactive species does not increase linearly with the nitrate concentration. Nonetheless, the exponential decrease in the nitrate photolysis rate with increasing nitrate concentration does not completely explain the other noticeable differences in the $k_{obs}$ results for aqueous aerosol-like vs. cloud/fog-like conditions. Firstly, the $k_{obs}$ values for MBO under aqueous aerosol-like conditions were factors of 1.2 to 5.1 higher than cloud/fog-like conditions. Reasons for MBO's higher $k_{obs}$ values under aqueous aerosol-like conditions are currently unknown. Secondly, in contrast to the insignificant effect that the ionic strength and/or sulfate concentration had on the reaction kinetics under the same pH conditions for cloud/fog-like conditions, the $k_{obs}$ values for the four GLVs were significantly higher at $I_{total} = 3.3$ M than at $I_{total} = 0.5$ M ($p < 0.05$) under the same pH conditions for aqueous aerosol-like conditions.

The higher $k_{obs}$ values at higher $I_{total}$ under the same pH conditions for aqueous aerosol-like conditions could be due to higher ·OH reactivities at higher ionic strength (Herrmann, 2003; Mekic and Gligorovski, 2021; Weller et al., 2010). Additionally, since (NH$_4$)$_2$SO$_4$ was used to control the ionic strengths of the solutions, sulfate photolysis likely contributed to the ionic strength/sulfate-dependent $k_{obs}$ trends for the aqueous aerosol-like conditions. Cope et al. (2022) showed that sulfur-containing radicals (e.g., SO$_4$·$^-$) were formed in ((NH$_4$)$_2$SO$_4$-containing concentrated solutions and aqueous aerosols when they were irradiated with UVB light or simulated sunlight. Even though the mechanism for the formation of sulfur-containing radicals from aqueous (NH$_4$)$_2$SO$_4$ photolysis remains unknown, the authors showed that the SO$_4$·$^-$ formed could easily react with various organic compounds in aqueous aerosols. The (NH$_4$)$_2$SO$_4$ concentrations used in this study (0.16 M and 1.09 M) were substantially lower than the (NH$_4$)$_2$SO$_4$ concentration used by Cope et al. (2022) (3.7 M). Nevertheless,



sulfate photolysis occurred under this study's aqueous aerosol-like conditions since the concentrations of the four GLVs decreased upon irradiation when only sulfate was present in the solutions (Figure S6), likely driven primarily by their reactions with $SO_4 \cdot^-$.

Our results clearly show that sulfur-containing radicals produced from sulfate photolysis can participate in aqueous reactions with GLVs in aqueous aerosols, but not in cloud and fog droplets. This could be due to the low hydration numbers in aqueous aerosols that would reduce the energy needed to produce sulfur-containing radicals from sulfate photolysis (Cope et al., 2022; Xu et al., 1998). As explained by Cope et al. (2022), a fully-solvated $SO_4^{2-}$ anion in dilute solutions has about 16 water molecules in its first solvation shell (Plumridge et al., 2000). The energy needed to detach an electron from $SO_4^{2-}$
$(H_2O)_n$ to produce $SO_4 \cdot^-(H_2O)_n$ decreases with the number of water molecules in its hydration shell, and electron detachment potentially occurs spontaneously at 3 water molecules (Pathak, 2014). Thus, the effective potential barrier for electron detachment of $SO_4^{2-}(H_2O)_4$ under concentrated conditions akin to aqueous aerosols is likely substantially lower compared to that under dilute conditions akin to cloud and fog droplets (Yang et al., 2002). Consequently, the likelihood of sulfur-containing radical production in concentrated aqueous aerosols would be substantially higher than that in diluted cloud and
fog droplets.

Comparisons of the $k_{obs}$ values obtained in the presence of sulfate and nitrate (Figure 4) vs. only sulfate (Figure S6) indicated that sulfate photolysis had a complex non-additive effect on the GLVs' reaction kinetics. Only approximately half of the $k_{obs}$ values measured in the presence of sulfate and nitrate were significantly higher ($p < 0.05$) than those measured in the presence of only sulfate under the same pH conditions. The non-additive effect that sulfate photolysis had on the GLVs'
reaction kinetics could be due to its mechanism coupling with the nitrate photolysis mechanism. For instance, $SO_4 \cdot^-$ could react with the $NO_3^-$ anion to form $NO_3 \cdot$ and the $SO_4^{2-}$ anion (De Semainville et al., 2007). However, since the mechanism for the formation of sulfur-containing radicals from sulfate photolysis remains unknown, we were unable to assess the extent by which the sulfate photolysis mechanism coupled with the nitrate photolysis mechanism under our experimental conditions. Furthermore, contributions of the $\cdot OH$, $SO_4 \cdot^-$, and $NO_3 \cdot$ reactions to the measured $k_{obs}$ would require knowledge of both the
reaction rate constants and concentrations of $\cdot OH$, $SO_4 \cdot^-$, and $NO_3 \cdot$. While the $SO_4 \cdot^-$ and $NO_3 \cdot$ concentrations in our study are not known, work by Sarang et al. (2021b) suggests that the rate constants for the reactions of GLVs with $\cdot OH$, $SO_4 \cdot^-$, and $NO_3 \cdot$ are on the orders of $10^9$ $M^{-1}$ $s^{-1}$, $10^8$ to $10^9$ $M^{-1}$ $s^{-1}$, and $10^7$ to $10^8$ $M^{-1}$ $s^{-1}$, respectively. However, the forementioned reaction rate constants were measured using dilute solutions with low ionic strengths (Sarang et al., 2021b), and it is unclear whether they could be extrapolated to aqueous aerosols which have high ionic strengths (Herrmann et al., 2015).

With the exception of tPtO at $I_{total} = 0.5$ M, the $k_{obs}$ values were higher at pH 5 than at pH 3 under the same ionic strength conditions, though these pH-dependent trends were not statistically significant ($p > 0.05$) in some instances (Tables S10 to S13). The increase in $k_{obs}$ with pH could potentially be due to the formation of $\cdot OH$, $SO_4 \cdot^-$, and $NO_3 \cdot$ from the coupled sulfate and nitrate photolysis mechanisms being pH-dependent, though this would require future studies to elucidate the





mechanisms. The $k_{obs}$ values measured in the presence of only sulfate did not have an obvious pH dependence (Figure S6),
which could be due to the four GLVs being pH-insensitive organic compounds. Cope et al. (2022) previously reported that
pH had substantial effects on the reactions of $SO_4\cdot^-$ with pH-sensitive organic compounds, but not on the reactions of $SO_4\cdot^-$
with pH-insensitive organic compounds.

### 3.2.2 AqSOA Mass Yields

Figure 5 shows the $Y_{SOA}$ values for the four GLVs measured at one GLV lifetime during irradiation in the presence
of nitrate at different pH (i.e., 3 vs. 5) and ionic strength (i.e., 0.5 M vs. 3.3 M). The $Y_{SOA}$ values measured under aqueous
aerosol-like conditions were substantially higher than those measured under cloud/fog-like conditions (Figure 3). The
enhanced aqSOA formation under aqueous aerosol-like conditions could be attributed to the higher concentrations of GLVs,
which were 100 times higher than those used under cloud/fog-like conditions. Consequently, the reaction of higher
concentrations of GLVs enhanced $RO_2\cdot$ and $RO\cdot$ combination reactions that led to oligomer formation.

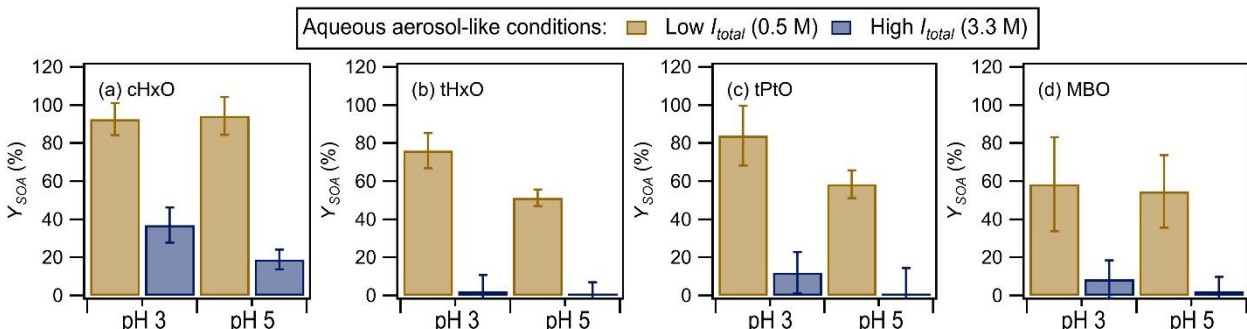

**Figure 5.** $Y_{SOA}$ values for the four GLVs at one GLV lifetime during nitrate-mediated photooxidation under cloud/fog-like
conditions. The error bars represent one standard deviation originating from triplicate experiments and measurements, and
include errors propagated from the standard deviations of the sulfate concentrations measured by ACSM. The ACSM-
measured organic signals for tPtO at pH 5 at $I_{total}$ = 3.3 M were very low, resulting in close to zero organic concentrations
and $Y_{SOA}$ values. Therefore, there was essentially no formation of low volatility products from this experiment, and the $Y_{SOA}$
value was essentially zero. Statistical analyses (student's $t$ test) on the differences in the $Y_{SOA}$ values are presented in Tables
S14 to S17.

Similar to the $Y_{SOA}$ measured under cloud/fog-like conditions (Figure 3), the $Y_{SOA}$ values for the four GLVs
generally decreased with increasing pH under the same ionic strength conditions, and with increasing ionic strength and
sulfate concentration under the same pH conditions, though these trends were not statistically significant ($p > 0.05$) in some
instances (Tables S14 to S17). The enhanced aqSOA formation at lower pH could be due to the formation of low volatility
products from acid-catalyzed reactions (e.g., hydration, polymerization, aldol condensation) (Ervens et al., 2011; Hallquist et





al., 2009; Jang et al., 2002), and/or the enhanced formation of low volatility organonitrates via the $RO_2\cdot + NO\cdot \rightarrow RONO_2$ pathway (Atkinson and Arey, 2003). Reduced aqSOA formation at higher ionic strength and sulfate concentration was likely

due to the enhancement of fragmentation pathways in the reactions of GLVs with sulfur-containing radicals formed from sulfate photolysis. For instance, $SO_4\cdot^-$ addition to C=C bonds to form higher molecular weight organosulfates is a minor channel compared to fragmentation pathways that form lower molecular weight products induced from electron transfer and other reactions by $SO_4\cdot^-$ (Ren et al., 2021). The higher concentrations of $SO_4\cdot^-$ formed from the photolysis of high concentrations of sulfate ($\geq 1085$ M) likely enhanced fragmentation pathways that led to the formation of lower molecular

weight products. Additionally, the higher $I_{total}$ conditions could have enhanced the partitioning of products to the gas phase due to the salting out effect (Peng and Wan, 1998).

Most noticeably, $Y_{SOA}$ values as high as 59 % were measured for MBO at $I_{total} = 0.5$ M under aqueous aerosol-like conditions, in contrast to the substantially lower $Y_{SOA}$ values measured under cloud/fog-like conditions ($\leq 9$ %). While this could be due to the enhancement of $RO_2\cdot$ and $RO\cdot$ combination reactions induced by the higher concentrations of MBO in

aqueous aerosol-like conditions, the formation of low volatility organosulfates induced by reactions involving sulfate could have contributed to the higher $Y_{SOA}$ values as well. The concentrations of sulfate used to control the ionic strength in aqueous aerosol-like conditions were up to 1861 times higher than those used in cloud/fog-like conditions (Table 1). Thus, the reaction of MBO with sulfur-containing radicals formed from sulfate photolysis likely played a significant role in aqSOA formation under aqueous aerosol-like conditions. Organosulfates (e.g., 2-hydroxy-2-methyl-4sulfate-3-butanone) were

previously identified as products from the reaction of MBO with $SO_4\cdot^-$ in the aqueous phase (Ren et al., 2021). Additionally, organosulfates could have been formed by acid-catalyzed reactions between sulfate and a MBO-derived epoxide (e.g., (3,3-dimethyloxiran-2-yl)methanol) formed from the ·OH reaction of MBO (Zhang et al., 2012). Nevertheless, increasing the sulfate concentration by 7 times to achieve $I_{total} = 3.3$ M led to substantial reductions in $Y_{SOA}$ (9 % and 2 %). This indicated that fragmentation pathways would eventually be enhanced in the reactions of GLVs with sulfur-containing radicals formed

from the photolysis of high concentrations of sulfate.

## 4 Conclusions and Implications

We investigated the nitrate-mediated photooxidation of four GLVs in dilute cloud/fog-like and concentrated aqueous aerosol-like conditions, focusing on the effects that pH, ionic strength, and sulfate on the reaction kinetics and aqSOA mass yields. Our results showed that the aqueous reaction medium conditions governed the effects that pH, ionic

strength, and sulfate had on the reaction kinetics and aqSOA mass yields. Under dilute cloud/fog-like conditions, the four GLVs had higher $k_{obs}$ at lower pH, which could be attributed to the pH-dependent formation of ·OH and other reactive species from nitrate photolysis. Ionic strength and sulfate had insignificant effects on $k_{obs}$. In contrast, under concentrated aqueous aerosol-like conditions, the four GLVs had higher $k_{obs}$ at higher pH, as well as higher $k_{obs}$ values at higher ionic





strength and sulfate concentration. Many of these differences could be attributed to sulfur-containing radicals produced from
sulfate photolysis participating in the reactions of GLVs under aqueous aerosol-like conditions, but not in cloud/fog-like
conditions. Under cloud/fog-like conditions where the sulfate concentrations were low, $k_{obs}$ was governed by the reactions of
GLVs with ·OH and other reactive species from nitrate photolysis. In contrast, the high sulfate concentrations in the aqueous
aerosol-like conditions enhanced the formation of sulfur-containing radicals from sulfate photolysis, which participated in
the reactions of GLVs. Higher $Y_{SOA}$ were measured under aqueous aerosol-like conditions, likely due to enhanced oligomer
formation from RO$_2$· and RO· combination reactions caused by the higher concentrations of GLVs reacted. Despite the
different effects that pH, ionic strength, and sulfate had on the reaction kinetics in cloud/fog-like vs. aqueous aerosol-like
conditions, similar $Y_{SOA}$ trends were observed for these two reaction conditions. Higher $Y_{SOA}$ was measured at lower pH,
which could be due to the enhanced formation of low volatility products from acid-catalyzed reactions and/or RO$_2$· + NO· →
RONO$_2$ reactions. Lower $Y_{SOA}$ was measured at higher ionic strength and sulfate concentration, which could be attributed to
the enhancement of fragmentation pathways in the reactions of GLVs with sulfur-containing radicals formed from sulfate
photolysis.

Overall, the results provide new insights into the aqueous photooxidation of GLVs in areas with substantial levels
of nitrate in cloud and fog droplets and aqueous aerosols. These insights are expected to be useful in modeling studies of the
atmospheric fates of GLVs and their contributions to the SOA budget. These insights built upon those provided by previous
studies that were conducted under dilute cloud/fog-like conditions and in the absence of inorganic salts (Richards-Henderson
et al., 2014; Richards-Henderson et al., 2015; Sarang et al., 2021b; Sarang et al., 2023). Results from this study highlight the
influences that nitrate and sulfate, the two main inorganic constituents in cloud and fog droplets and aqueous aerosols in
most regions, can have on the aqueous photooxidation of GLVs. Additionally, the magnitudes of their influences depend on
the aqueous reaction medium (i.e., dilute cloud and fog droplets vs. concentrated aqueous aerosols) in which the reactions
occur in.

Our study also highlights many questions about the sulfate photolysis mechanism that need to be addressed in
future studies. These include the mechanism for the formation of sulfur-containing radicals from aqueous (NH$_4$)$_2$SO$_4$
photolysis, and how the sulfate photolysis mechanism can couple with the nitrate photolysis mechanism to affect the
formation of reactive species including ·OH. Figure S3 showed that the [·OH]$_{ss}$ decreased with increasing sulfate
concentration under dilute cloud/fog-like conditions (Figure S3), though the magnitude of the decrease depended on the pH.
Additional studies are needed to elucidate how the presence of sulfate will affect the formation of reactive species under
different conditions (e.g., pH, ionic strength, aqueous reaction medium). While not investigated in this study due to our
inability to completely remove inorganic salts prior to UPLC-MS analysis, we hypothesize that [·OH]$_{ss}$ will likely similarly
decrease with increasing sulfate concentration under concentrated aqueous aerosol-like conditions. Additionally, even
though this study focuses on the aqueous photooxidation of GLVs, it is likely that nitrate, sulfate, and the aqueous reaction
medium will influence the aqueous photooxidation of other water-soluble organic compounds as well. More importantly, the



manner in which nitrate and sulfate influence the reaction kinetics and aqSOA formation will not only depend on the aqueous reaction medium in which the reactions occur in, but also whether the water-soluble organic compound is pH-sensitive or pH-insensitive (Cope et al., 2022; Lyu et al., 2023; Yang et al., 2023). The forementioned factors need to be
considered in future studies on the photooxidation of water-soluble organic compounds in different atmospheric aqueous phases.

       There are several caveats that should be noted. First, we were unable to completely distinguish the effects of sulfate and ionic strength on the aqueous photooxidation of GLVs since $H_2SO_4$ and $(NH_4)_2SO_4$ were used to control both the pH and ionic strength of the solutions. Second, the effects of only two ionic strength conditions, 0.5 M and 3.3 M, were investigated
in experiments simulating aqueous aerosols. However, ionic strengths in atmospheric aqueous aerosols span a large range and can reach 45 M (Herrmann et al., 2015; Volkamer et al., 2007). Future studies could consider using a chemically inert inorganic salt (e.g., sodium perchlorate (Mekic et al., 2018a)) to control the ionic strength of solutions and investigate reactions in aqueous aerosols with very high ionic strengths. Third, many of our conclusions regarding key reaction pathways were drawn based on $Y_{SOA}$ measurements performed using an ACSM due to our inability to completely remove
inorganic salts from experimental samples before UPLC-MS analysis. Future studies should consider using alternative analytical methods that are not adversely impacted by inorganic salts (e.g., gas chromatography-mass spectrometry (Sarang et al., 2023) and nuclear magnetic resonance (Ren et al., 2021)) to identify prominent products and reaction pathways, though the detection of oligomers is still expected to be analytically challenging.

**Data availability**

The data used in this publication are available to the community and can either be accessed on request to the corresponding author or online at: https://doi.org/10.5281/zenodo.14829906 (Nah et al., 2025).

**Author contributions**

YL: Conceptualization, Investigation, and Writing – original draft & editing. TJ: Investigation, Writing – review & editing. RM, MKEC, TZ, SY, CKW, and YG: Investigation. YQ: Writing – review & editing. TN: Conceptualization, Writing –
review & editing, Supervision. All authors reviewed the manuscript and agreed to the final version.

**Competing interests**

At least one of the (co-)authors is a member of the editorial board of Atmospheric Chemistry and Physics. The authors have no other competing interests to declare.

**Financial support**



The work described in this paper was supported by a grant from the Research Grants Council of the Hong Kong Special
Administrative Region, China (project numbers 11303720 and 11303321).

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
