# Peer review of "Roles of pH, ionic strength, and sulfate in the aqueous nitratemediated photooxidation of green leaf volatiles"

_EGUsphere, 2025_

## Author Response (AR3)

We thank both referees for their thoughtful, constructive, and encouraging comments on our manuscript. We greatly appreciate the time and effort they have invested in reviewing our work. Their feedback has been highly valuable in improving the clarity, completeness, and overall quality of the paper.

In the following, we provide a point-by-point response to each of the referees' comments. For clarity, the referee comments are shown in *italics*, and our responses are given in regular (non-italicized) text, any quotation and addition of new sentences in the revised manuscript are shown in **bold**. The line numbers in referees' comments refer to the lines in the original preprint and the those in our response refer to the lines in the revised manuscript.

**Comments to Referee #1**

***General comments from referee***

*This manuscript describes measurements that aim to understand the SOA-forming capacity of green leaf volatiles (GLV) that react with the products of the aqueous phase photolysis of nitrate. Specifically, the work describes measurements of the overall reaction rate constants ($k_{obs}$) for 4 specific GLVs via high-resolution time-of-flight electrospray ionization mass spectrometer (HR-ToF-ESI) and separate experiments to determine the SOA yields. In order to investigate both cloud/fog and aerosol-like conditions, both the ionic strength and pH of the solutions were varied. Importantly, ammonium sulfate was used to control the ionic strength of the solutions, which led to complications in the interpretation of the results. Under dilute cloud/fog-like conditions, the four GLVs had higher $k_{obs}$ at lower pH, which could be attributed to the pH-dependent formation of OH and other reactive species from nitrate photolysis. Ionic strength and sulfate had insignificant effects on $k_{obs}$. In contrast, under concentrated aqueous aerosol-like conditions, the four GLVs had higher $k_{obs}$ at higher pH, as well as higher $k_{obs}$ values at higher ionic strength and sulfate concentration. These effects are explained by the expected nitrate photolysis-initiated processes as well as the unexpected role of sulfate-related oxidation processes. Higher SOA yields under both cloud/fog and aerosol-like conditions were observed at lower pH, which was attributed to acid-catalyzed accretion reactions.*

*Because of the importance of the study in helping to refine the formation mechanisms of SOA from such precursors as GLVs, this work will be of interest to general readers of EGUsphere. The experiments are rationally designed, thoroughly analyzed, and the manuscript is generally well written. However, the work is difficult to assess as it seems that it was designed as a careful study of the nitrate photolysis-initiated processes, but that design was compromised by the presence of unanticipated sulfate photolysis mechanisms. The authors admit that the ionic strength dependence of the nitrate photolysis-initiated processes needs to be reinvestigated with a non-sulfate species. The finding of sulfate photolysis-related processes is very important and worth reporting but is likewise complicated by the concurrent nitrate photolysis mechanism. Therefore, it is quite obvious to the reader that new experiments should be designed that isolate the nitrate and sulfate photolysis processes. Nonetheless, even though the work was not able to achieve its original goals of determining a rigorous quantitative understanding of the nitrate photolysis-related processes, it is still valuable as a qualitative outline of the combined importance of the nitrate and sulfate photolysis pathways.*

*There are several items that should be addressed in a revised version of the manuscript:*

1. *Line 235: It would have been relatively to test this hypothesis with a separate experiment that generated OH exclusively. Is there a reason this was not done?*

**Response:** The SOA mass yields ($Y_{SOA}$) from the reaction of ·OH and GLVs have already been studied by Richards-Henderson et al. (2014) utilizing $H_2O_2$ photolysis as the ·OH photochemical precursor. The $Y_{SOA}$ values reported by Richards-Henderson et al. (2014) for cHxO and MBO are very close to those measured in our study. Additionally, we showed in a previous study (Lyu et al., 2023) that reaction with ·OH is the main contributor to the degradation kinetics of non-photolyzable organic compounds during aqueous nitrate-mediated photooxidation. Given the results from these studies, we concluded that the decays of the GLVs and aqSOA formation are governed mostly by their reactions with ·OH without conducting separate experiments to test this hypothesis. To remove any confusion, we have made the following changes to the revised manuscript:

**Page 8 line 203: "Reaction with ·OH was also shown to be the main contributor to the reaction kinetics of other non-photolyzable organic compounds (e.g., formic acid, glycolic**

**acid) during aqueous nitrate-mediated photooxidation (Lyu et al., 2023). While it is possible that sulfur-containing radicals and other reactive species were formed from the photolysis of $(NH_4)_2SO_4$ (Table S4), their effects on $k_{obs}$ are small due to their low concentrations under diluted cloud/fog-like conditions (Cope et al., 2022). Additionally, the $k_{obs}$ values measured under illumination in control experiments conducted in the presence of sulfate only were not statistically different ($p > 0.05$) from the $k_{obs}$ values measured under illumination in control experiments conducted in the absence of nitrate and sulfate ("light only" experiments). Thus, the decays of the GLVs were likely governed mostly by their reactions with ·OH, though minor contributions from their reactions with reactive species other than ·OH cannot be discounted."**

2. *Line 246: Why couldn't the inorganic salts be separated before analysis?*

**Response:** Solid phase extraction (SPE) usually can efficiently remove inorganic salts from non-polar organic compounds with reversed phased sorbent such as HLB (hydrophilic-lipophilic balanced sorbent). However, in our study, the investigated organic compounds are GLVs that have hydroxyl functional groups (-OH). The -OH functional group makes the GLV less retainable during the "water washing" step in a typical SPE protocol. We attempted to remove the inorganic salts with HLB and MAX cartridges, but it was difficult to balance between removing salts and retaining the GLVs. Additionally, the investigated GLVs all have low molecular weights and low UPLC-MS ionization efficiencies. This resulted in us being unable to separate the inorganic salts from GLVs in our samples and still obtain sufficient signals above the baseline during UPLC-MS analyses.

3. *Line 377: This is a very out of date set of references for acid catalyzed SOA processes. I suggest adding:*

   *Epoxides: Cooke et al. ES&T, 58, 10675-10684, 2024*

   *Acetals: Presberg et al. ACS Earth and Space Chem., 8, 1634-1645, 2024*

   *Oligomers: Maben et al., Environ Sci Process Impacts, 25, 214-228, 2023*

**Response:** The references of Jang et al. (2002) and Hallquist et al. (2009) have been removed and replaced with the three references recommended. The following changes have been made in the revised manuscript:

**Page 11 Line 279: "Some of these carbonyls could have undergone acid-catalyzed reactions (e.g., hydration, polymerization, aldol condensation) to form low volatility products (Ervens et al., 2011; Maben and Ziemann, 2023; Presberg et al., 2024; Cooke et al., 2024)."**

**Page 15 Line 390: "The enhanced aqSOA formation at lower pH could be due to the formation of low volatility products from acid-catalyzed reactions (e.g., hydration, polymerization, aldol condensation) (Ervens et al., 2011; Maben and Ziemann, 2023; Presberg et al., 2024; Cooke et al., 2024), and/or the enhanced formation of low volatility organonitrates via the $RO_2\cdot + NO\cdot \rightarrow RONO_2$ pathway (Atkinson and Arey, 2003)."**

4. *Line 379: Why would there be enhanced formation of organonitrates from $RO_2$ + NO at high ionic strength?*

**Response:** Figure 5 shows that the $Y_{SOA}$ values for the four GLVs generally decreased with increasing pH under the same ionic strength conditions, and with increasing ionic strength and sulfate concentration under the same pH conditions.

Based on the referee's comment, we assume that they are referring to the sentence in the original manuscript: "*The enhanced aqSOA formation at lower pH could be due to the formation of low volatility products from acid-catalyzed reactions (e.g., hydration, polymerization, aldol condensation) (Ervens et al., 2011; Maben and Ziemann, 2023; Presberg et al., 2024; Cooke et al., 2024) and/or the enhanced formation of low volatility organonitrates via the $RO_2\cdot + NO\cdot \rightarrow RONO_2$ pathway (Atkinson and Arey, 2003).*"

This sentence refers to our proposed explanation as to why enhanced formation of aqSOA was observed at lower pH. It does not state that there was enhanced formation of organonitrates from $RO_2$ + NO at high ionic strength.

We subsequently explained in the sentences after the aforementioned sentence as to why reduced aqSOA formation was observed at higher ionic strength and sulfate concentration:

"*Reduced aqSOA formation at higher ionic strength and sulfate concentration was likely due to the enhancement of fragmentation pathways in the reactions of GLVs with sulfur-containing radicals formed from sulfate photolysis. For instance, $SO_4^{\cdot-}$ addition to C=C bonds to form higher molecular weight organosulfates is a minor channel compared to fragmentation pathways that form lower molecular weight products induced from electron transfer and other reactions by $SO_4^{\cdot-}$ (Ren et al., 2021). The higher concentrations of $SO_4^{\cdot-}$ formed from the photolysis of high concentrations of sulfate (≥ 1085 M) likely enhanced fragmentation pathways that led to the formation of lower molecular weight products. Additionally, the higher $I_{total}$ conditions could have enhanced the partitioning of products to the gas phase due to the salting out effect (Peng and Wan, 1998).*"

5. *Technical comments:*

- *Line 64: typo "ideal"*
- *Line 154: extraneous "the" between "from" and "before"*
- *Line 313: typo in subscript for ionic strength "total"*

**Response:** The following changes have been made in the revised manuscript:

**Page 2 Line 62: "Under the high ionic strength conditions in aqueous aerosols, substantial ion association occurs, which will affect the activity coefficients of organic compounds, resulting in reactions occurring under non-ideal conditions (Herrmann, 2003)."**

**Page 6 Line 158: "Solid phase extraction (SPE) using SPE cartridges (Oasis MAX, 60 mg, 3 cc, 60 µm, Waters; Bond PPL Elut, 200 mg, 3 mL, 125 µm, Agilent) was performed to remove inorganic salts from samples before UPLC-MS analysis."**

**Page 13 Line 327: "The higher $k_{obs}$ values at higher $I_{total}$ under…"**

**Comments to Referee #2**

***General comments from referee***

*The present manuscript explores the % secondary organic aerosol (SOA) contribution of green leaf volatiles (GLVs) resulting via nitrate mediated aqueous photooxidation pathway in presence of sulfate in varied reaction conditions. To do so, the authors determined the first order decay kinetics and SOA yields of the selected GLVs i.e., cis-3-hexen-1-ol (cHxO), trans-2-hexen-1-ol (tHxO), trans-2-penten-1-ol (tPtO), 2-methyl-3-buten-2-ol (MBO) resulting from the nitrate mediated photooxidation. They investigated the role of pH, ionic strength, and sulfate on these investigated reactions. The pseudo first order reaction rate constants ($k_{obs}$) for the selected GLVs were determined by following the GLV concentrations at different time intervals using ultrahigh-performance liquid chromatography coupled to a photodiode array detector (UPLC-PDA). While the %SOA yield was determined using the highly sophisticated Aerosol Chemical Speciation Monitor (ACSM). The authors applied different liquid aerosol conditions i.e., cloud/fog droplets-like and aqueous aerosol-like to determine the effect of pH, ionic strength and sulfate on the investigated reactions. The use of ammonium sulfate to control the ionic strength led clearly to the undetermined effect on these reactions. Briefly, presence of sulfate as a salt can first act as a precursor source of sulfate radicals in the system (being produced during nitrate photolysis); secondly as an additional pathway of reaction with nitrates resulting into nitrate radical and sulfate anion.*

*In cloud/fog-like conditions $k_{obs}$ for the GLVs are relatively higher at lower pH=3 than at pH=5, while ionic strength had statistically insignificant effect. However, the %SOA yield was observed to be higher for the low pH=3 and low ionic strength (I=0.002 M). While with varying ionic strength the $k_{obs}$ remains practically unchanged, the %yield does not. This is possibly due to the shift in increased side reaction of GLVs with increase in the concentrations of other radicals such as $SO_4^{-}\cdot$ anion radicals in the system at higher ionic strength. This shift does not significantly influence the overall $k_{obs}$ while it completely changes the pathway of reaction resulting into more volatile fragments from GLV-sulfate reactions and hence the lower %SOA yield.*

*In aqueous aerosol-like conditions the concentrations of nitrate and GLVs were set to be 100 times higher, while concentrations of salts i.e., ammonium sulfate to control ionic strength was 100 –*

*1000 times higher in different scenarios listed in Table 1. The $k_{obs}$ here, increased with increasing ionic strength, while the pH had insignificant effect. The increased concentration of nitrates in the system results in exponential decrease in the nitrate photolysis rate and thereby decreased concentration of OH radicals (based on literature cited), resulting into little lower impact of pH on $k_{obs}$ in this case. However, increasing the ionic strength may increase the presence of $SO_4^{-}\cdot$ anion radicals resulting into increased $k_{obs}$. The increased aqSOA at lower pH could be attributed to the formation of higher amount of low volatility product from acid catalyzed reactions. Additionally, the reactions at higher concentrations of GLVs in aqueous aerosol-like conditions results into higher amount of $RO_2\cdot$ and $RO\cdot$ combination reactions, possibly resulting into oligomers.*

*Since the presented work in the manuscript builds on to fill the existing gap in knowledge concerning the under investigated role of GLVs in the atmospheric reaction and SOA contribution, it will be highly valuable to the readers of EGUsphere and in general the atmospheric science community. The approach to determine the first order kinetics and %SOA yield is analytically sound and rational. The authors have tried to address the effect of pH, ionic strength and sulfate on the nitrate mediated aqueous-phase reactions of GLVs. But the gap in understanding remains, due to the additional experiments with ·OH clearly required to resolve the same. The authors admits and state to investigate the role of sulfate in future studies using inert salts such as sodium perchlorate instead of ammonium sulfate. Additionally, aqueous phase reactions of these GLVs should be studied with ·OH ($H_2O_2$) and HONO, respectively to evaluate the overall contribution of respective pathways of reactions with ·OH, $NO_3\cdot$, $NO_2\cdot$, NO· and $SO_4^{-}\cdot$. It is understandable that these studies are deemed important however, are beyond the scope of the present work as it could make the study quite exhaustive. Despite the listed short comings, the manuscript thoroughly examines the kinetics and resulting SOA yield. The overall quality of the results and their interpretation within the study is of high scientific quality and significance.*

*Scientific comments*

*The manuscript holds potential for acceptance and would serve valuable for EGU Journal Atmospheric Chemistry and Physics in relatively-less investigated role of GLVs where very little to moderate is known in comparison to the other biogenic compounds such as isoprene, monoterpenes, and sesquiterpenes. The present listed issues need to be addressed before acceptance:*

1. *“Line 133: The apparent or pseudo first order rate constants are usually determined in the conditions where one of the reactants is always in excess. However, the first order rate constants fit well according to equation 2 as stated. (Include one of these exemplary plots in SI).”*

**Response:** As requested, we have added plots that show the fits to GLV decays in the SI:

**Page 6 Line 135: “All the decays of the GLVs followed apparent first order reaction kinetics reasonably well (Figures S3 and S4), thus they were fitted with the following equation:”**

[Figure]

**Figure S3.** Decays of the GLVs in the absence (“Light only”) and presence of nitrate and sulfate under cloud/fog-like conditions (Table 1). The error bars represent one standard deviation originating from triplicate experiments and triplicate measurements at each reaction time. The $k_{obs}$ at different pH and ionic strengths were corrected for MBO under cloud/fog-like conditions.

[Figure]

**Figure S4.** Decays of the GLVs in the absence ("Light only") and presence of nitrate and sulfate under aqueous aerosol-like conditions (Table 1). The error bars represent one standard deviation originating from triplicate experiments and triplicate measurements at each reaction time. The $k_{obs}$ at different pH and ionic strengths were corrected for MBO under aqueous aerosol-like conditions.

2. *"Line 201: The decay of GLVs is governed by their reactions with ·OH, however the experiments were never tested against ·OH which is potentially a control experiment in the case. Can authors state the reason why it was not done in this case?"*

**Response:** Richards-Henderson et al. (2014) previously investigated the kinetics of and aqSOA yields from reactions of a series of GLVs with ·OH using $H_2O_2$ photolysis as the ·OH photochemical precursor. Two of the GLVs (cHxO and MBO) used in this nitrate-mediated

photooxidation study were used in the study by Richards-Henderson et al. (2014). The $Y_{SOA}$ values reported by Richards-Henderson et al. (2014) for cHxO and MBO are very close to those measured in our study. Additionally, we showed in our previous work (Lyu et al., 2023) that reaction with ·OH is the main contributor to the degradation kinetics of non-photolyzable organic compounds during aqueous nitrate-mediated photooxidation. Given the results from these prior studies, we concluded that the decays of the GLVs and aqSOA formation are governed mostly by their reactions with ·OH without conducting separate control experiments using $H_2O_2$ photolysis to avoid unnecessary duplicated experiments. To remove any confusion, we have made the following changes to the revised manuscript:

**Page 8 line 203: "Reaction with ·OH was also shown to be the main contributor to the reaction kinetics of other non-photolyzable organic compounds (e.g., formic acid, glycolic acid) during aqueous nitrate-mediated photooxidation (Lyu et al., 2023). While it is possible that sulfur-containing radicals and other reactive species were formed from the photolysis of $(NH_4)_2SO_4$ (Table S4), their effects on $k_{obs}$ are small due to their low concentrations under diluted cloud/fog-like conditions (Cope et al., 2022). Additionally, the $k_{obs}$ values measured under illumination in control experiments conducted in the presence of sulfate only were not statistically different ($p > 0.05$) from the $k_{obs}$ values measured under illumination in control experiments conducted in the absence of nitrate and sulfate ("light only" experiments). Thus, the decays of the GLVs were likely governed mostly by their reactions with ·OH, though minor contributions from their reactions with reactive species other than ·OH cannot be discounted."**

3. *"Line 101 in the SI: In Table S1, although the sulfate photolysis and resulting side reactions seems to significantly influence the reactions, they are omitted from the presentation. It would be useful, if authors can present the list of the set of reactions in different reaction conditions within SI:"*

    1) *List of reactions possibly governing experimentally studied Cloud/fog like conditions and aqueous aerosol conditions.*

    2) *Control experiments (in presence and absence of sulfate/nitrate/OH, respectively).*

**Response:** The full set of reactions resulting from sulfate photolysis is currently unknown. This includes the formation of sulfur containing radicals from the photolysis of $(NH_4)_2SO_4$ even though both our study and Cope et al. (2022) have provided strong evidence for their formation. Nevertheless, we have added a list of reactions pathways hypothesized to be associated with the aqueous photolysis of sulfate compiled from the current literature into the revised manuscript:

**Table S4.** List of reactions pathways hypothesized to be associated with the aqueous photolysis of sulfate compiled from the literature (Cope et al., 2022; De Semainville et al., 2007; Herrmann et al., 1999). Note that the mechanisms behind the formation of sulfur containing radicals from the aqueous photolysis of $(NH_4)_2SO_4$ are still unknown.

| No. | Reactions |
|---|---|
| 1 | $SO_4^{2-} + H^+ \leftrightarrows HSO_4^-$ |
| 2 | $\bullet OH + HSO_4^- \rightarrow SO_4\bullet^- + H_2O$ |
| 3 | $SO_4\bullet^- + SO_4\bullet^- \rightarrow S_2O_8^{2-}$ |
| 4 | $SO_4\bullet^- + HO_2\bullet \rightarrow SO_4^{2-} + H^+ + O_2$ |
| 5 | $SO_4\bullet^- + O_2^- \rightarrow SO_4^{2-} + O_2$ |
| 6 | $SO_4\bullet^- + OH^- \rightarrow SO_4^{2-} + \bullet OH$ |
| 7 | $SO_4\bullet^- + H_2O \rightarrow SO_4^{2-} + H^+ + \bullet OH$ |
| 8 | $S_2O_8^{2-} + h\nu \rightarrow 2SO_4\bullet^-$ |
| 9 | $SO_4\bullet^- + NO_3^- \rightarrow SO_4^{2-} + NO_3\bullet$ |

4. *"Line 110 in the SI: Which case of recorded molar absorptivity for $NH_4NO_3$ is presented in Figure S2? Was the molar absorptivity of $NH_4NO_3$ in presence of GLVs, $(NH_4)_2SO_4$, and $H_2SO_4$ within the standard deviation of the one in the absence?"*

**Response:** The molar absorptivity calculated for $NH_4NO_3$ as presented in Figure S2 was recorded in absence of any organic compound or inorganic salt. The caption of Figure S2 has been revised in the revised manuscript. We have also added a figure showing how the molar absorptivity of $NH_4NO_3$ changes when GLVs, $(NH_4)_2SO_4$, and $H_2SO_4$ are added into the solution. The following changes have been made in the revised manuscript:

**Page 4 Line 114: "Only the addition of tPto and tHxO had significant effects on the molar absorptivity of $NH_4NO_3$ (Figure S2b), enhancing the peak molar absorptivity of $NH_4NO_3$ by approximately 1.3 and 1.5 times, respectively."**

[Figure]

**Figure S2.** (a) Photon flux inside the Rayonet photoreactor under our experimental conditions (black solid line), and (b) molar absorptivities (ε) of the solutions of 25 mM $NH_4NO_3$ (black dotted line) and 25 mM $NH_4NO_3$ mixed with 1085 mM $(NH_4)_2SO_4$ (red solid line), 0.5 mM $H_2SO_4$ (orange solid line), 10 mM cHxO (green solid line), 10 mM MB (blue solid line), 10 mM tHxO (pink solid line), and 10 mM tPto (grey solid line). Also shown are the error bars of the peak molar absorptivities of the different solutions. The error bars represent one standard deviation originating

from triplicate absorption measurements. Only the addition of tPto and tHxO were found to have significant effects on the peak molar absorptivities of $NH_4NO_3$ ($p < 0.05$).

5. *"Line 117: Sec 2.2 Photochemistry Experiments: The below mentioned comments should be addressed to increase the quality of the work and make easier for future studies to replicate the same."*

- *What does open cylindrical quartz tubes mean here? If the reactor is kept open aren't GLVs susceptible to escape as they are moderately volatile? See Henry's constant. And it is expected some GLVs (as stated for MBO) might be lost as well. Please, include in SI, the concentration time-series for the dark control vs the photo experiments for all GLVs.*

**Response:** The quartz tubes were open to air. We observed some loss in the GLVs during illumination in control experiments conducted in the absence of nitrate and sulfate, but not during dark control experiments. These losses were attributed to evaporation due to the volatilities of the GLVs and the temperature within the reactor (30 °C during photochemistry and "light only" control experiments compared to 23 °C during dark control experiments). MBO had the largest loss, likely due to its highest vapor pressure. To remove any confusion, we have made the following changes in the revised manuscript:

**Page 6 line 140: "No loss in GLVs was observed in dark control experiments conducted in the absence and presence of nitrate and sulfate. During illumination in control experiments conducted in the absence of nitrate and sulfate ("light only" experiments), only MBO had significant loss under cloud/fog-like conditions, whereas all four GLVs had significant losses under aqueous aerosol-like conditions (Figures S3 and S4). The four GLVs were not expected to undergo direct photolysis as they do not absorb light significantly at wavelengths larger than 280 nm (Richards-Henderson et al., 2014; Sarang et al., 2021a), as demonstrated in Figures S5 and S6. Thus, the observed losses under illumination in control experiments conducted in the absence of nitrate and sulfate could be due to evaporation, with MBO having the largest losses due to its higher vapor pressure ($3.08 \times 10^{-2}$ atm) compared to the other three GLVs (cHxO: $1.23 \times 10^{-3}$ atm, tHxO: $1.20 \times 10^{-3}$ atm, and tPtO: $3.46 \times 10^{-3}$ atm) based on estimations using EPI Suite™ (U.S. EPA, 2024). The $k_{obs}$ values measured for the**

**GLVs decays in nitrate-mediated photooxidation experiments were subsequently corrected by subtracting the loss rates from control experiments conducted in the absence of nitrate and sulfate ("light only" experiments)."**

- *What was the total volume of reactor and reaction solution?*

**Response:** This information has been added into the revised manuscript:

**Page 5 line 124: "The volume of each quartz tube was around 15 mL. The quartz tubes were filled with 10 mL and 1 mL of solutions during experiments simulating cloud/fog-like and aqueous aerosol-like conditions, respectively. Aliquots of 1 mL and 0.1 mL were extracted from the illuminated solutions at different reaction times for offline chemical analysis during experiments simulating cloud/fog-like and aqueous aerosol-like conditions, respectively."**

- *What is the total reaction time in each case?*

**Response:** The total reaction times ranged from 25 minutes to 150 min depending on how quickly the GLVs decayed under different experimental conditions. This information can be obtained from Figures S3 and S4 that have been added into the SI. Please refer to our reply to comment 1.

- *Line 123: How much was the temperature deviation within the reactor from 30°C? As rate constants can be sensitive to the temperature change. Is it thermostatic with something other than cooling fan and how was it monitored through the reaction?*

**Response:** The temperature inside the photoreactor was periodically monitored using a thermometer. The temperature remained constant at 30 °C with little fluctuation (± 0.2 °C) throughout all the photochemistry experiments. The temperature inside the photoreactor was kept constant using only the cooling fan located at the bottom of the photoreactor.

- *Why was 30°C chosen as a temperature? What are the thoughts of author on temperature dependence of these reactions? Isn't that necessary for future studies?*

**Response:** The temperature was not chosen intentionally. The temperature was the net result of the balance between the heat released from the 16 illuminated UVB lamps and the cooling effect provided by the fan located at the bottom of the photoreactor. We hypothesize that temperature will likely influence the kinetics of the reactions. However, since the temperature inside the photoreactor was constant at 30 °C throughout all the photochemistry experiments in this study, we do not expect temperature to affect our results, which focused on the influence of pH, ionic strength, and sulfate on the nitrate-mediated photooxidation of GLVs. Temperature should be controlled in future studies to limit its effect on the kinetics of reactions.

- *Line 125: At different reaction time? How frequently was it sampled to determine the kinetics? Was it same for all experiments? If not, what was the number of aliquots taken out in each case respectively. In case of AqSOA yield aliquots were extracted from the illuminated solutions at one GLV lifetime (Include these lifetime values in SI at least for readers reference).*

**Response:** Information on the reaction times and number of data points for each experimental condition can be obtained from Figures S3 and S4 that have been added into the SI. Please refer to our reply to comment 1. As requested, the $k_{obs}$ values and respective one lifetime for the four GLVs under different conditions can be found in Tables S2 and S3 added into the SI. The following changes have been made in the revised manuscript:

**Page 7 Line 163: "Aliquots of 10 mL and 1 mL were extracted from the illuminated solutions at one GLV lifetime (i.e., $\tau = \frac{1}{k_{obs}}$, when 37 % of the initial concentration of the GLV remained) in experiments simulating cloud/fog-like and aqueous aerosol-like conditions, respectively (Tables S2 and S3)."**

**Page 7 Line 178: "The subscripts $\tau$ and 0 indicate the sample solutions obtained at one GLV lifetime (Tables S2 and S3) and before illumination, respectively."**

**Table S2.** List of $k_{obs}$ and one lifetime (i.e., $\tau = \frac{1}{k_{obs}}$, when 37 % of the initial concentration of the GLV remained) of GLVs during nitrate-mediated photooxidation under cloud/fog-like conditions (Table 1).

| GLVs | cHxO | | tHxO | | tPtO | | MBO | |
|---|---|---|---|---|---|---|---|---|
| | $k_{obs}$ (s$^{-1}$) | $\tau$ (min) | $k_{obs}$ (s$^{-1}$) | $\tau$ (min) | $k_{obs}$ (s$^{-1}$) | $\tau$ (min) | $k_{obs}$ (s$^{-1}$) | $\tau$ (min) |
| pH 3 $I_{total}$ = 0.002 M | 6.02×10$^{-5}$ | 277 | 6.98×10$^{-5}$ | 239 | 1.06×10$^{-4}$ | 158 | 3.23×10$^{-4}$ | 52 |
| pH 3 $I_{total}$ = 0.02 M | 5.61×10$^{-5}$ | 297 | 6.74×10$^{-5}$ | 247 | 1.04×10$^{-4}$ | 161 | 3.41×10$^{-4}$ | 49 |
| pH 5 $I_{total}$ = 0.002 M | 5.5×10$^{-5}$ | 301 | 5.69×10$^{-5}$ | 293 | 8.22×10$^{-5}$ | 203 | 2.76×10$^{-4}$ | 60 |
| pH 5 $I_{total}$ = 0.02 M | 5.6×10$^{-5}$ | 298 | 5.82×10$^{-5}$ | 287 | 8.23×10$^{-5}$ | 203 | 2.92×10$^{-4}$ | 57 |

**Table S3.** List of $k_{obs}$ and one lifetime (i.e., $\tau = \frac{1}{k_{obs}}$, when 37 % of the initial concentration of the GLV remained) of GLVs during nitrate-mediated photooxidation under aqueous aerosol-like conditions (Table 1).

| GLVs | cHxO | | tHxO | | tPtO | | MBO | |
|---|---|---|---|---|---|---|---|---|
| | $k_{obs}$ (s$^{-1}$) | $\tau$ (min) | $k_{obs}$ (s$^{-1}$) | $\tau$ (min) | $k_{obs}$ (s$^{-1}$) | $\tau$ (min) | $k_{obs}$ (s$^{-1}$) | $\tau$ (min) |
| pH 3 $I_{total}$ = 0.5 M | 7.91×10$^{-6}$ | 2108 | 1.00×10$^{-5}$ | 1663 | 1.74×10$^{-5}$ | 959 | 3.86×10$^{-4}$ | 43 |
| pH 3 $I_{total}$ = 3.3 M | 2.74×10$^{-5}$ | 608 | 3.82×10$^{-5}$ | 437 | 3.04×10$^{-5}$ | 548 | 1.05×10$^{-3}$ | 16 |
| pH 5 $I_{total}$ = 0.5 M | 8.55×10$^{-6}$ | 1949 | 1.08×10$^{-5}$ | 1538 | 1.60×10$^{-5}$ | 1045 | 4.65×10$^{-4}$ | 36 |
| pH 5 $I_{total}$ = 3.3 M | 2.78×10$^{-5}$ | 599 | 4.32×10$^{-5}$ | 386 | 3.36×10$^{-5}$ | 497 | 1.50×10$^{-3}$ | 11 |

- *Line 135: "Only MBO showed some loss". Was it quantified? Any specific reason why concentration-time series are not included within SI?*

**Response:** The time series in the decays of the GLVs have been added into the SI. Please refer to our reply to comment 1. We clarified the instances where GLVs had significant losses during illumination in control experiments conducted in the absence of nitrate and sulfate in the revised manuscript:

**Page 6 line 140: "No loss in GLVs was observed in dark control experiments conducted in the absence and presence of nitrate and sulfate. During illumination in control experiments conducted in the absence of nitrate and sulfate ("light only" experiments), only MBO had significant loss under cloud/fog-like conditions, whereas all four GLVs had significant losses under aqueous aerosol-like conditions (Figures S3 and S4). The four GLVs were not expected to undergo direct photolysis as they do not absorb light significantly at wavelengths larger than 280 nm (Richards-Henderson et al., 2014; Sarang et al., 2021a), as demonstrated in Figures S5 and S6. Thus, the observed losses under illumination in control experiments conducted in the absence of nitrate and sulfate could be due to evaporation, with MBO having the largest losses due to its higher vapor pressure ($3.08 \times 10^{-2}$ atm) compared to the other three GLVs (cHxO: $1.23 \times 10^{-3}$ atm, tHxO: $1.20 \times 10^{-3}$ atm, and tPtO: $3.46 \times 10^{-3}$ atm) based on estimations using EPI Suite™ (U.S. EPA, 2024). The $k_{obs}$ values measured for the GLVs decays in nitrate-mediated photooxidation experiments were subsequently corrected by subtracting the loss rates from control experiments conducted in the absence of nitrate and sulfate ("light only" experiments)."**

- *Line 139: Sarang et al., 2023 highlights "HEXAL, in contrast, absorbs light in the range of 290–400 nm (ε = 51–0.6 $M^{-1}$ $cm^{-1}$, c.f. SI Section 1) and was therefore isomerized to the Z-isomer within irradiation experiments". It is unreasonable that no loss of this GLV occurred during illumination. Please provide the molar absorptivity of GLVs recorded at 311 nm, which is a peak of photon flux for the presently used reactor. Thus, concentration time-series of GLVs for all cases: experiments and control are recommended to be included in SI.*

**Response:** The HEXAL used in Sarang et al. (2023) is (*E*)-2-hexen-1-al, which is a light-absorbing GLV species due to it having an aldehyde functional group. In our case, the four GLVs only contain one C=C double bond and one hydroxyl function group, and thus are not light-absorbing GLV species. Insignificant light absorption by the non-aldehyde containing GLVs was also observed by Richard-Henderson et al. (2014, 2015) and reviewed by Sarang et al. (2023). The time series in the decays of the GLVs have been added into the SI. Please refer to our reply to comment 1. The absorbance spectra of the GLVs have been added as Figures S5 and S6 in the SI:

[Figure]

**Figure S5.** Light absorption spectra of the four GLVs at different ionic strengths under cloud/fog-like condition. The GLV concentrations were set to 0.1 mM, and the ionic strength of the solutions were adjusted with only $H_2SO_4$ and $(NH_4)_2SO_4$. The absorbances of all the solutions were weak in the spectral region of the light output in the Rayonet photoreactor (Figure S2).

[Figure]

**Figure S6.** Light absorption spectra of the four GLVs at different ionic strengths under cloud/fog-like condition. The GLV concentrations were set to 0.1 mM, and the ionic strength of the solutions were adjusted with only $H_2SO_4$ and $(NH_4)_2SO_4$. The slightly increased absorption from 275 to 325 nm could be due to the additions of large amounts of $(NH_4)_4SO_4$ (Cope et al., 2022). In general, the absorbances of all the solutions were weak in the spectral region of the light output in the Rayonet photoreactor (Figure S2).

6. *"Line 199: "Other reactive species produced during nitrate photolysis (e.g., hydroperoxide radicals and superoxide ions are also expected to have lower reactivities compared to ·OH". Mention sulfate radicals, as they are also present in the system."*

**Response:** For cloud/fog-like conditions, the $k_{obs}$ values measured under illumination in control experiments conducted in the presence of sulfate only were not statistically different ($p > 0.05$) from the $k_{obs}$ values measured under illumination in control experiments conducted in the absence of nitrate and sulfate ("light only" experiments). This indicated that sulfur containing radicals

originating from $(NH_4)_2SO_4$ photolysis had little effect on the degradation kinetics of the GLVs under cloud/fog-like conditions. The following changes have been made in the revised manuscript:

**Page 8 Line 205: "While it is possible that sulfur containing radicals and other reactive species were formed from the photolysis of $(NH_4)_2SO_4$ (Table S4), their effects on $k_{obs}$ are small due to their low concentrations under diluted cloud/fog-like conditions (Cope et al., 2022). Additionally, the $k_{obs}$ values measured under illumination in control experiments conducted in the presence of sulfate only were not statistically different ($p > 0.05$) from the $k_{obs}$ values measured under illumination in control experiments conducted in the absence of nitrate and sulfate ("light only" experiments)."**

7. *"Line 295: The steady state concentrations of OH could not be determined experimentally in conditions of higher ionic strengths, however, is it possible to do the same using some kinetic modelling to provide approximate value. See (OH estimation model) in Otto et al., The Journal of Physical Chemistry A 2017 121 (34), 6460-6470.*

   *This would serve as a contrast to understand the role of OH even better between cloud/fog-like vs. aqueous aerosol-like conditions. Also, it will strengthen the statement in Line 305."*

**Response:** While COPASI is a powerful tool for simulating biochemical networks, it requires detailed reaction mechanisms along with precise kinetic parameters to accurately model species concentrations. At present, the full set of reactions resulting from sulfate photolysis is currently unknown, including the formation of sulfur containing radicals from the photolysis of $(NH_4)_2SO_4$. The sulfate photolysis mechanism can couple with the nitrate photolysis mechanism to influence ·OH formation (De Semainville et al., 2007), but the extend of this coupling is not currently not known. These issues prevent us from using COPASI to simulation ·OH formation in our system.

Instead, we refined our SPE methodology and successfully removed the inorganic salts from the solutions used to simulate aqueous aerosol-like conditions, allowing us to estimate $[\cdot OH]_{ss}$ for aqueous aerosol-like conditions. Similar pH-dependent trends are observed for ·OH formation under aqueous aerosol-like conditions and cloud/fog-like conditions, with the estimated $[\cdot OH]_{ss}$

decreasing when pH increases from 3 to 5. The estimated [·OH]$_{ss}$ for the cloud/fog-like and aqueous aerosol-like conditions have been added into the SI:

[Figure]

**Figure S7.** Estimated [·OH]$_{ss}$ in nitrate-mediated photooxidation experiments under (a) cloud/fog-like, and (b) aqueous aerosol-like conditions. These values were obtained from a separate set of experiments (i.e., GLVs were not present in the solutions) using benzoic acid (10 µM) as the ·OH probe compound and measuring the formation of p-hydroxybenzoic acid from the reaction of ·OH with BA (Lyu et al., 2023; Yang et al., 2021; Yang et al., 2023). The error bars represent one standard deviation originating from triplicate experiments and triplicate measurements. For the low $(NH_4)_2SO_4$ concentration conditions (red bars), 0.135 mM and 0.583 mM of $(NH_4)_2SO_4$ was added into the solutions for pH 3 and 5, respectively, for cloud/fog-like conditions, whereas 158 mM of $(NH_4)_2SO_4$ was added into the solutions for both pH 3 and 5 for aqueous aerosol-like

conditions (Table 1). For the high $(NH_4)_2SO_4$ concentration conditions (blue bars), 6.135 mM and 6.580 mM of $(NH_4)_2SO_4$ was added into the solutions for pH 3 and 5, respectively, for cloud/fog-like conditions, whereas 1085 mM was added into the solutions for both pH 3 and 5 for aqueous aerosol-like conditions (Table 1). At present, it is unclear why the $[\cdot OH]_{ss}$ increased with $(NH_4)_2SO_4$ concentration at pH 5 under aqueous aerosol-like conditions.

8. *"Line 301 – 305: The resulting decrease of the first order reaction rate constant can be attributed to the lower rate of reactions of GLVs with sulfate ($10^8$ L mol$^{-1}$ s$^{-1}$) than OH ($10^9$ L mol$^{-1}$ s$^{-1}$). The rate constants are mentioned in Line 347, however, is it possible that these differences in the second order rate constant values could also explain partially the decrease in $k_{obs}$ under aqueous aerosol-like conditions with clearly higher sulfate present in the system.*

   *The authors could list and compare the literature available rate constants for the GLVs against ·OH, $SO_4^-$·and $NO_3$· within the SI and conclude in a sentence within the manuscript for the ease of understanding the concept and scientific clarity of the reader."*

**Response:** The referee is correct that the lower $k_{obs}$ values measured for these three GLVs under aqueous aerosol-like conditions could also be due to sulfur-containing radicals reacting with GLVs. We have added this possibility to the revised manuscript. As requested, we have added Table S5 which shows the second-order reaction rate constants for cHxO and MBO with ·OH reported by Richard-Henderson et al. (2014), and the second-order reaction rate constants for 1-peten-3-ol, *cis*-2-hexen-1-ol, and *trans*-2-hexen-1-al with ·OH, $SO_4^-$·and $NO_3$· reported by Sarang et al. (2021) into the SI. The following changes have been made in the revised manuscript:

**Page 13 Line 316: "Additionally, sulfur-containing radicals and reactive species produced from sulfate photolysis are expected to contribute significantly to the degradation of the GLVs under aqueous aerosol-like conditions due to the high concentration of $(NH_4)_2SO_4$ in the solutions (Cope et al., 2022). Work by Sarang et al. (2021b) suggests that the rate constants for the reactions of GLVs with $SO_4^{\cdot-}$ is about 1 order of magnitude lower than those of their reactions with ·OH (Table S5)."**

**Page 14 Line 358: "While the $SO_4\cdot^-$ and $NO_3\cdot$ concentrations in our study are not known, work by Richards-Henderson et al. (2014) and Sarang et al. (2021b) suggests that the rate constants for the reactions of GLVs with ·OH, $SO_4\cdot^-$, and $NO_3\cdot$ are on the orders of $10^9$ $M^{-1}$ $s^{-1}$, $10^8$ to $10^9$ $M^{-1}$ $s^{-1}$, and $10^7$ to $10^8$ $M^{-1}$ $s^{-1}$, respectively (Table S5)."**

**Table S4.** Previously reported second-order reaction rate constants for the GLVs against ·OH, $SO_4\cdot^-$, and $NO_3\cdot$.

| GLVs | Oxidant | Rate constant ($\times 10^{-9}$ $M^{-1}$ $s^{-1}$) | Temp. (K) | pH | Reference |
|---|---|---|---|---|---|
| cHxO | ·OH | 5.1 ± 0.8 | 298 | 3.1 | (Richards-Henderson et al., 2014) |
| | | 5.3 ± 0.3 | | 5.4 | |
| | | 5.3 ± 0.2 | | 6.9 | |
| MBO | ·OH | 7.5 ± 1.4 | 298 | 3.1 | (Richards-Henderson et al., 2014) |
| | | 8.0 ± 0.6 | | 5.4 | |
| | | 7.3 ± 0.7 | | 6.9 | |
| 1-peten-3-ol | ·OH | 6.3 ± 0.1 | 298 | 7 | (Sarang et al., 2021) |
| | $SO_4\cdot^-$ | 0.94 ± 0.10 | | | |
| | $NO_3\cdot$ | 0.15 ± 0.015 | | | |
| *cis*-2-hexen-1-ol | ·OH | 6.7 ± 0.3 | 298 | 7 | (Sarang et al., 2021) |
| | $SO_4\cdot^-$ | 2.5 ± 0.3 | | | |
| | $NO_3\cdot$ | 0.84 ± 0.23 | | | |
| *trans*-2-hexen-1-al | ·OH | 4.8 ± 0.3 | 298 | 7 | (Sarang et al., 2021) |
| | $SO_4\cdot^-$ | 0.48 ± 0.02 | | | |
| | $NO_3\cdot$ | 0.03 ± 0.07 | | | |

*9. Technical and typographical comments:*

*1) Line 29; The sentence should be corrected. Present form results into interpretation that other that Isoprene and monoterpenes, the rest/remaining comes solely from GLVs.*

*2) For better clarity state the approximate %BVOC contribution of GLVs to the remaining half (citing literature).*

**Response:** We thank the referee for drawing out attention to this ambiguous statement. The current emission of total GLVs is not well constrained since atmospheric models such as MEGAN do not treat GLVs as a separate group. Many GLV species and their sources are still not well documented.

Thus, it would be difficult to give an estimate of the contribution of GLVs to the total BVOCs. To remove any confusion, the following changes have been made in the revised manuscript:

**Page 1 Line 28 "Isoprene and monoterpenes comprise more than half of the total annual BVOC emissions (Sindelarova et al., 2014). Since green leaf volatiles (GLVs) comprise a comparatively smaller fraction of total BOVCs, their chemical processes have received far less attention compared to isoprene and monoterpene. GLVs are $C_5$ to $C_6$ unsaturated organic compounds with aldehyde, alcohol, or ester functional groups (Sarang et al., 2021a)."**

*3) Line 64: typo "non-deal" should be corrected.*

**Response:** The typo has been corrected in the revised manuscript:

**Page 2 Line 62: "Under the high ionic strength conditions in aqueous aerosols, substantial ion association occurs, which will affect the activity coefficients of organic compounds, resulting in reactions occurring under non-ideal conditions (Herrmann, 2003)."**

*4) Line 80: Figure 1 structure names: cis and trans could be in italics*

**Response:** Figure 1 has been replaced with the figure shown below:

[Figure]

**Figure 1. The four model GLVs used in this study.**

*5) Line 103: typo/grammatical error "to study pH the effects".*

**Response:** The typo has been corrected in the revised manuscript:

**Page 4 Line 103: "The pH of unbuffered solutions (no addition of $H_2SO_4$) were close to 5, and it was selected as the higher bound to study the pH effects on the nitrate-mediated photooxidation of GLVs."**

*6) Line 106: The pH (i.e., pH 3 vs. 5, Table 1)*

**Response:** The following changes have been made in the revised manuscript:

**Page 4 Line 107: "The pH (i.e., pH 3 vs. 5, Table 1) used in this study fall within the ranges for cloud and fog droplets and aqueous aerosols (Herrmann et al., 2015; Pye et al., 2020; Tilgner et al., 2021)."**

*7) Line 111: add (Table 1) in brackets to direct the reader for better clarity.*

**Response:** The following changes have been made in the revised manuscript:

**Page 4 Line 111: "The ionic strengths used in this study, i.e., 0.002 M vs. 0.02 M for cloud/fog-like conditions and 0.5 M vs. 3.3 M for aqueous aerosol-like conditions (Table 1), fall within the ranges for ionic strengths for clouds/fog droplets and continental aerosols (Herrmann et al., 2015)."**

*8) Line 142: Correct reference text format (U.S. EPA, 2024)*

**Response:** This have been corrected in the revised manuscript:

**Page 6 Line 148: "(U.S. EPA, 2024)."**

*9) Line 154: Incorrect article/preposition usage "to remove inorganic salts from the before UPLC-MS analysis"*

**Response:** The following changes have been made in the revised manuscript:

**Page 6 Line 157: "Solid phase extraction (SPE) using two different types of SPE cartridges (Oasis MAX, 60 mg, 3 cc, 60 µm, Waters; Bond PPL Elut, 200 mg, 3 mL, 125 µm, Agilent) was performed to remove inorganic salts from samples before UPLC-MS analysis."**

We thank both referees and the editor for their thoughtful, constructive, and encouraging comments on our manuscript. We greatly appreciate the time and effort they have invested in reviewing our work. Their feedback has been highly valuable in improving the clarity, completeness, and overall quality of the paper.

In the following, we provide a point-by-point response to the editor's minor comment. For clarity, the editor's comments are shown in *italics*, and our responses are given in regular (non-italicized) text, any quotation and addition of new sentences in the revised manuscript are shown in **bold**.

***General comments from the editor***

*There is one remaining issue to address prior to accepting your manuscript for publication. In Figure S2 in the SI, the UV light spectrum for the Rayonet photoreactor is shown. This has a significant tail < 280 nm and even a prominent small peak around 250 nm. Photons < 280 nm are not present in the troposphere, and usually efforts are made in such environmental photochemistry experiments to filter out photons < 280 nm. These higher energy photons can excite chromophores and drive photochemistry that is not accessible in the true UVB region > 280 nm.*

*This is not a fatal flaw but the presence of UVC light in these experiments and its possible interferences through driving photochemistry not possible in the troposphere or at the Earth's surface will have to be made clear in the main manuscript. Going forward I would suggest looking for other UVB lamps that do not emit < 280 nm.*

**Response:** The editor's comment is noted. We have revised the stated wavelength range in the revised manuscript:

Page 5 line 122: **"The photon flux in the photoreactor ranged from 250 to 400 nm and peaked at 311 nm (Figure S2a)."**